# On Uniform Convergence
# and Low-Norm Interpolation Learning

**Lijia Zhou**
University of Chicago
zlj@uchicago.edu

**Danica J. Sutherland**
TTI-Chicago
danica@ttic.edu

**Nathan Srebro**
TTI-Chicago
nati@ttic.edu

## Abstract

We consider an underdetermined noisy linear regression model where the minimum-norm interpolating predictor is known to be consistent, and ask: can uniform convergence in a norm ball, or at least (following Nagarajan and Kolter) the subset of a norm ball that the algorithm selects on a typical input set, explain this success? We show that uniformly bounding the difference between empirical and population errors cannot show any learning in the norm ball, and cannot show consistency for any set, even one depending on the exact algorithm and distribution. But we argue we can explain the consistency of the minimal-norm interpolator with a slightly weaker, yet standard, notion: uniform convergence *of zero-error predictors* in a norm ball. We use this to bound the generalization error of low- (but not minimal-) norm interpolating predictors.

## 1 Introduction

In the past several years, it has become empirically clear that – contrary to traditional intuition – it is possible for models which exactly interpolate noisy training data to reliably generalize well on practical problems, especially in deep learning [7, 25, 34]. We refer to this phenomenon as "interpolation learning." It is closely related to the (re-)discovery of the "double descent" phenomenon [1, 5, 23, 29], where many models first improve as their size is increased, then get much worse around the point where they can first interpolate the data, and then improve again as they become more and more overparametrized. Understanding interpolation learning, therefore, seems to be a key step on the path towards better theoretical understanding of the successes of deep learning.

We now know of a few settings where interpolating models can be shown to generalize well [4, 8]. In particular, significant recent attention has been paid to the minimum-norm linear interpolator ("ridgeless" regression) in certain high-dimensional linear regression regimes [3, 6, 14, 21]. This setting is of particular interest not only because it is reasonably accessible to study while exhibiting many of the surprising properties of more complex models, but also because this predictor is the same one found by (stochastic) gradient descent initialized at the origin, and so it seems plausible that its properties may generalize to more complex settings. Much is now understood about the properties of the minimum-norm interpolator for (sub-)Gaussian data, including necessary and sufficient conditions for its consistency. This line of inquiry has proved quite fertile for extensions to related settings and further results [2, 13, 15, 18, 20].

One striking feature of this body of work is that none of it is based on the core workhorse of learning theory, *uniform convergence*; most instead uses various tools, mostly from random matrix theory, to directly analyze the generalization error of a particular predictor. Indeed, some have argued that uniform convergence is unlikely to be able to explain interpolation learning; for instance, Mikhail Belkin has said[1] that "there are no [uniform generalization] bounds" with constants tight enough to

explain interpolation learning, "and no reason they should exist." Meanwhile, Nagarajan and Kolter [22] have also raised significant questions about the ability of uniform convergence arguments to explain learning in certain high-dimensional regimes. Perhaps, then, it is time to wholly abandon uniform convergence in favor of other tools.

We connect these two avenues of work by studying uniform convergence in a particular over-parametrized linear regression problem (Section 2) where the minimal-norm interpolator is consistent. We prove that, indeed, uniform convergence bounds based on predictor norm cannot show *any* learning in this setting (Theorem 3.2). We also prove, following Nagarajan and Kolter, that *no* uniform convergence bound can show consistency (Theorem 3.3), not only for the minimal-norm interpolator but even for a wide variety of natural interpolation algorithms.

Yet, even in this setting where the situation looks bleak, we need not abandon uniform convergence entirely. One option would be sidestep the negative results by considering uniform convergence not of our predictor, but of a surrogate separately shown to be not too different [24]. We instead demonstrate that it is possible to show uniform convergence of our predictor directly if we allow ourselves a slightly weaker notion of uniform convergence, one long in common use in realizable PAC analyses: uniform convergence *for predictors with zero error*. Such a bound would be implied by, for example, "optimistic rates" [30], although existing results are not tight enough to show consistency in our setting. Instead we prove (Theorem 4.1) that a tight version of this notion of uniform convergence *does* hold in our setting for low-norm predictors. Our result exactly characterizes the asymptotic worst-case generalization gap for predictors of a given norm via a novel analysis based on strong duality of a particular non-convex problem, and show that while neither having a low norm nor interpolation is sufficient for generalization in our setting, the combination is. By doing so, not only do we prove consistency of the minimal-norm interpolator with a uniform convergence-type argument, we also provide new insight about the behavior of interpolation learning for solutions with low but not minimal norm.

## 2 Problem setting

We begin with a standard linear regression setup, with Gaussian data and errors. Take i.i.d. observations $(x_1, y_1), ..., (x_n, y_n) \sim \mathcal{D}^n$, where the joint distribution $\mathcal{D}$ is given by

**A** $x \in \mathbb{R}^p$ is drawn from $\mathcal{N}(0, \Sigma)$, with $\Sigma \succ 0$, and $\epsilon \in \mathbb{R}$ is independently $\mathcal{N}(0, \sigma^2)$. There is some fixed $w^* \in \mathbb{R}^p$ such that $y = \langle w^*, x \rangle + \epsilon$.

We consider a "junk features" setting, where $x$ decomposes into "signal" and "junk" components, and analysis of interpolation learning is particularly appealing:

**B** In Setting A, let $\Sigma = \begin{bmatrix} I_{d_S} & 0_{d_S \times d_J} \\ 0_{d_J \times d_S} & \frac{\lambda_n}{d_J} I_{d_J} \end{bmatrix}$ where $d_S, d_J$ satisfy $d_S + d_J = p$, and $\lambda_n > 0$.

In other words, we can write $x = (x_S, x_J)$, where $x_S \sim \mathcal{N}(0, I_{d_S})$ and $x_J \sim \mathcal{N}(0, \frac{\lambda_n}{d_J} I_{d_J})$. Further, the label depends only on $x_S$: $w^* = (w_S^*, 0_{d_J})$ with $w_S^* \in \mathbb{R}^{d_S}$.

Let $Y \in \mathbb{R}^n$ be the vector of responses, $X \in \mathbb{R}^{n \times p}$ the design matrix and $E \in \mathbb{R}^n$ the residual vector, so $Y = Xw^* + E$. The sample covariance is $\hat{\Sigma} = \frac{1}{n} X^\mathsf{T} X$. The population and empirical risks are, respectively,

$$L_{\mathcal{D}}(w) = \mathbb{E}_{(x,y) \sim \mathcal{D}}[(y - \langle w, x \rangle)^2] = L_{\mathcal{D}}(w^*) + \|w - w^*\|_{\Sigma}^2$$

$$L_{\mathbf{S}}(w) = \frac{1}{n} \|Y - Xw\|^2 = L_{\mathbf{S}}(w^*) + \|w - w^*\|_{\hat{\Sigma}}^2 - \frac{2}{n} \langle X^\mathsf{T} E, w - w^* \rangle, \tag{1}$$

where $\|x\|_A = \sqrt{x^\mathsf{T} A x}$ denotes the Mahalonobis norm, and $L_{\mathcal{D}}(w^*) = \mathbb{E}\,\epsilon^2 = \sigma^2$.

We will focus on the regime where $d_S$ is fixed, and $d_J \to \infty$ for *finite* values of $n$, e.g. $\lim_{n \to \infty} \lim_{d_J \to \infty} L_{\mathcal{D}}(\hat{w})$. This setting enables relatively easy calculation of many quantities of interest, and can recover many interesting behaviors of overparametrized interpolation, including consistency and the double descent phenomenon.

We will be primarily concerned with the behavior of the minimal-norm interpolator,

$$\hat{w}_{MN} = \operatorname*{arg\,min}_{w \in \mathbb{R}^p \text{ s.t. } Xw = Y} \|w\|_2^2 = X^\mathsf{T} (XX^\mathsf{T})^{-1} Y. \tag{2}$$

This predictor is in fact consistent in Setting **B** when $\lambda_n = o(n)$ and we consider $d_J \to \infty$ for each $n$. We here use a slightly broader notion of *consistency* than is traditional [e.g. in 28]: we mean that

$$\mathbb{E}\left[L_{\mathcal{D}}(\hat{w}_{MN}) - L_{\mathcal{D}}(w^*)\right] \to 0$$

for our *sequence* of learning problems in the given asymptotic regime. Specifically:

**Proposition 2.1.** *In Setting **B** with $\lambda_n = o(n)$,*

$$\lim_{n \to \infty} \lim_{d_J \to \infty} \mathbb{E}\left[L_{\mathcal{D}}(\hat{w}_{MN}) - L_{\mathcal{D}}(w^*)\right] = 0.$$

The proof follows from Lemmas 2.2 and 2.3, which establish first – because the setting was designed exactly to make this true[2] – that $\hat{w}_{MN}$ becomes equivalent to ridge regression on the signal part of $X$ with regularization weight $\lambda_n$, and then that ridge regression is consistent in this setting.

Writing $X = (X_S, X_J)$ with $X_S \in \mathbb{R}^{n \times d_S}$ and $X_J \in \mathbb{R}^{n \times d_J}$, the ridge regression estimate on the signal components with tuning parameter $\lambda$ is given by

$$\hat{w}_\lambda = \arg\min_{w \in \mathbb{R}^p} \|Y - X_S w\|^2 + \lambda\|w\|^2$$
$$= (X_S^\mathsf{T} X_S + \lambda I_{d_S})^{-1} X_S^\mathsf{T} Y = X_S^\mathsf{T}(X_S X_S^\mathsf{T} + \lambda I_n)^{-1} Y.$$

**Lemma 2.2.** *In Setting **B**,* $\lim_{d_J \to \infty} \mathbb{E}[L_{\mathcal{D}}(\hat{w}_{MN})] = \mathbb{E}[L_{\mathcal{D}}(\hat{w}_{\lambda_n})]$ *for any $n$.*

*Proof.* By the strong law of large numbers, we have that $X_J X_J^\mathsf{T} = \lambda_n \frac{Z_J Z_J^\mathsf{T}}{d_J}$ converges almost surely to $\lambda_n I_n$. Writing $\hat{w}_{MN} = (\hat{w}_{MN,S}, \hat{w}_{MN,J})$, we can easily verify that

- $\hat{w}_{MN,S} = X_S^\mathsf{T}(X_S X_S^\mathsf{T} + X_J X_J^\mathsf{T})^{-1} Y \overset{a.s.}{\to} \hat{w}_{\lambda_n}$ by the continuous mapping theorem.
- $\hat{w}_{MN,J} = X_J^\mathsf{T}(X_S X_S^\mathsf{T} + X_J X_J^\mathsf{T})^{-1} Y$. Drawing a new $x_J \sim \mathcal{N}(0, \frac{\lambda_n}{d_J} I_{d_J})$, $X_J x_J \overset{a.s.}{\to} 0_n$ and so $\langle \hat{w}_{MN,J}, x_J \rangle \overset{a.s.}{\to} 0$.

This implies that for any fixed $x$, $\langle \hat{w}_{MN}, x \rangle \overset{a.s.}{\to} \langle \hat{w}_{\lambda_n}, x_S \rangle$, and hence via continuity we have that $(\langle \hat{w}_{MN}, x \rangle - y)^2 \overset{a.s.}{\to} (\langle \hat{w}_{\lambda_n}, x \rangle - y)^2$. Taking expectations over $(x, y)$ to get $L_{\mathcal{D}}$ and then over the training set, then exchanging the limit with each expectation,[3] we obtain the desired result. $\square$

**Lemma 2.3.** *In Setting **B**, if $\lambda_n = o(n)$, then* $\lim_{n \to \infty} \mathbb{E}\left[L_{\mathcal{D}}(\hat{w}_{\lambda_n}) - L_{\mathcal{D}}(w^*)\right] = 0.$

The proof, as for all the following results, is in the appendix. Taking $\lambda_n = o(n)$ ensures the bias due to regularization is negligible; the minimax-optimal scaling would be $\lambda_n \propto \sqrt{n}$ [11].

**Relationship to previous settings**   The results of Bartlett et al. [3] apply to our setting, also showing consistency of $\hat{w}_{MN}$. Although they do not require $p \to \infty$ for finite $n$ as we study, their results show that consistency of $\hat{w}_{MN}$ is only possible when the effective $p$ grows much faster than $n$. Muthukumar et al. [21] showed that *no* interpolation method can be consistent in Setting **A** for $p = \mathcal{O}(n)$; we re-derive this (simple) result in Proposition 4.3, since it will also be important for our purposes.

Hastie et al. [14] and various follow-ups, on the other hand, employ the standard asymptotic regime of random matrix theory, where $n/p \to \gamma \in (0, \infty)$, mostly focusing on $\Sigma = I$. Although no interpolator can achieve consistency here, they exactly evaluate $\lim_{(n,d) \to \infty} L_{\mathcal{D}}(\hat{w}_{MN})$. The setting of Belkin et al. [6] is related, with general $(n, p)$ but again with $\Sigma = I$, where $\hat{w}_{MN}$ is not consistent.

# 3 Uniform convergence

We now know, via Proposition 2.1, that $\hat{w}_{MN}$ is consistent in this setting. Could we have discovered this fact directly via uniform convergence? Typically, we would find some class $\mathcal{W}_{n,\delta}$ such that $\Pr(\hat{w}_{MN} \in \mathcal{W}_{n,\delta}) \geq 1 - \delta$, and bound the generalization gap

$$\Pr\left(\sup_{w \in \mathcal{W}_{n,\delta}} L_{\mathcal{D}}(w) - L_{\mathbf{S}}(w) \leq \epsilon_{\mathcal{W}}(n, \delta)\right) \geq 1 - \delta. \tag{3}$$

As $L_{\mathbf{S}}(\hat{w}_{MN}) = 0$, this would directly provide an upper bound on $L_{\mathcal{D}}(\hat{w}_{MN})$ with probability $1 - 2\delta$.

## 3.1 Uniform convergence over norm balls

Our first thought would likely be to find some high-probability upper bound $B_{n,\delta}$ on $\|\hat{w}_{MN}\|$, and take $\mathcal{W}_{n,\delta} = \{w \in \mathbb{R}^p : \|w\| \leq B_{n,\delta}\}$. We can get a rough asymptotic estimate for $B_{n,\delta}$ based on the following, since $\|\hat{w}_{MN}\| = \mathcal{O}_P\left(\sqrt{\mathbb{E}\|\hat{w}_{MN}\|^2}\right)$ by Markov's inequality.

**Proposition 3.1.** *As $n \to \infty$ in Setting B, if $\lambda_n$ is both $o(n)$ and $\omega(1)$, then*

$$\lim_{d_J \to \infty} \mathbb{E}\|\hat{w}_{MN}\|^2 = \frac{\sigma^2 n}{\lambda_n} + \mathcal{O}(1) \quad and \quad \lim_{d_J \to \infty} \frac{(\mathbb{E}\|\hat{w}_{MN}\|^2)(\mathbb{E}\|x\|^2)}{n} = \sigma^2 + o(1).$$

We could then find $\epsilon_{\mathcal{W}}(n, \delta)$ by studying the Rademacher complexity, given by

$$\mathfrak{R}_n(\mathcal{W}_B) = \mathbb{E}_{\mathbf{S}} \, \mathbb{E}_{\sigma \sim \mathrm{Unif}(\pm 1)^n} \sup_{w : \|w\| \leq B} \frac{1}{n} \sum_{i=1}^{n} \sigma_i \langle w, x^{(i)} \rangle \leq \sqrt{\frac{1}{n} B^2 \, \mathbb{E}\|x\|^2};$$

thus Proposition 3.1 gives us that $\mathfrak{R}_n(\mathcal{W}_{\sqrt{\mathbb{E}\|\hat{w}_{MN}\|^2}}) \leq \sigma + o(1)$.

Standard Rademacher bounds are for Lipschitz losses, which the squared loss is not. If we let $T_n$ be a uniform upper bound on all the labels and $Q_n$ on all the predictions, however, the absolute value of the derivative of the squared loss is at most $2|\hat{y} - y| \leq 2(Q_n + T_n)$, and so we can treat it as $2(Q_n + T_n)$-Lipschitz with high probability. We then obtain in the setting of Proposition 3.1 that

$$\sup_{\|w\|^2 \leq \mathbb{E}\|\hat{w}_{MN}\|^2} L_{\mathcal{D}}(w) - L_{\mathbf{S}}(w) \leq 4(Q_n + T_n)\left(\sigma + \mathcal{O}_P\left(\frac{1}{\sqrt{n}}\right)\right). \tag{4}$$

To show consistency, we need a bound exactly approaching $\sigma^2$ as $n \to \infty$, i.e. $Q_n + T_n \to \frac{1}{4}\sigma$. But in fact, each of $Q_n$ and $T_n$ diverge to $\infty$ as $n \to \infty$, because we have more and more chances to see a large value. Thus for $n \to \infty$, (4) says nothing at all.

Now, the path to (4) was potentially quite loose, particularly in the Lipschitz step; perhaps, then, we could simply put more effort in to obtain the bound we want. This is not the case: balls which are big enough to contain $\hat{w}_{MN}$ also contain predictors with unbounded generalization gaps as $n \to \infty$.

**Theorem 3.2.** *In Setting B, if $\lambda_n = o(n)$ then*

$$\lim_{n \to \infty} \lim_{d_J \to \infty} \mathbb{E}\left[\sup_{\|w\| \leq \|\hat{w}_{MN}\|} |L_{\mathcal{D}}(w) - L_{\mathbf{S}}(w)|\right] = \infty.$$

*Proof sketch.* Proposition B.2 shows that the gap is at least $\|\Sigma - \hat{\Sigma}\|(\|\hat{w}_{MN}\| - \|w^*\|)^2 + o(1)$ using (1) and then aligning $w - w^*$ with $\Sigma - \hat{\Sigma}$. By Proposition 3.1, $(\|\hat{w}_{MN}\| - \|w^*\|)^2$ grows like $n/\lambda_n$. Now $\|\Sigma - \hat{\Sigma}\|$ goes to 0, but only at the rate of $\sqrt{\lambda_n/n}$ [17], so the product grows as $\sqrt{n/\lambda_n}$. $\square$

Proposition B.2 also gives a lower bound for $\mathbb{E}\left[\sup_{\|w\| \leq \|\hat{w}_{MN}\|} L_{\mathcal{D}}(w) - L_{\mathbf{S}}(w)\right]$, the one-sided generalization gap, based on the algebraically largest eigenvalue of $\Sigma - \hat{\Sigma}$ rather than the operator norm. We expect that this eigenvalue should asymptotically behave similarly to the operator norm, and hence the one-sided generalization gap should also diverge.

Norm balls around $w^*$, rather than the origin, fare no better; they would merely remove the asymptotically irrelevant $\|w^*\|$ term from the result of Proposition B.2.

## 3.2 Uniform convergence over algorithm- and distribution-dependent hypothesis classes

Choosing $\mathcal{W}_{n,\delta}$ as a Euclidean norm ball, then, cannot yield the result we want (or, indeed, any meaningful result at all for large $n$). But a norm ball doesn't fully capture everything we know about $\hat{w}_{MN}$: for instance, we know that its norm is not likely to be very small. Perhaps taking a shell rather than a ball would help? Following Nagarajan and Kolter [22], we show that in fact, *no* choice of $\mathcal{W}_{n,\delta}$ can demonstrate consistency using the most common two-sided uniform convergence bounds.

Specifically, let $\mathcal{S}_{n,\delta}$ be a set of typical training examples $\mathbf{S} = (X, Y)$ such that $\Pr(\mathbf{S} \in \mathcal{S}_{n,\delta}) \geq 1 - \delta$, let $\mathcal{A}(X, Y)$ be any learning algorithm, and then take the class of typical outputs of $\mathcal{A}$, $\mathcal{W}_{n,\delta}^{\mathcal{A}} = \{\mathcal{A}(X, Y) : (X, Y) \in \mathcal{S}_{n,\delta}\}$. (Clearly, no bound based on $\mathcal{S}_{n,\delta}$ could choose a smaller $\mathcal{W}_{n,\delta}$.) The *tightest algorithm-dependent uniform convergence bound* [22] is then

$$\sup_{\mathbf{S} \in \mathcal{S}_{n,\delta}} \sup_{w \in \mathcal{W}_{n,\delta}^{\mathcal{A}}} |L_{\mathcal{D}}(w) - L_{\mathbf{S}}(w)| \leq \epsilon_{\mathcal{A}}^{\mathcal{D}}(n, \delta), \tag{5}$$

$$\text{implying } \Pr\left(|L_{\mathcal{D}}(\mathcal{A}(X, y)) - L_{\mathbf{S}}(\mathcal{A}(X, y))| \leq \epsilon_{\mathcal{A}}^{\mathcal{D}}(n, \delta)\right) \geq 1 - \delta.$$

In interpolation learning, where $L_{\mathbf{S}}$ is zero, we need $\lim_{n \to \infty} \epsilon_{\mathcal{A}}^{\mathcal{D}}(n, \delta) = \sigma^2$ to obtain consistency. Nagarajan and Kolter show that in a particular high-dimensional linear classification setting, stochastic gradient descent has 0 asymptotic loss, but $\epsilon_{\mathcal{A}}^{\mathcal{D}}(n, \delta)$ must be nearly 1 for any $\mathcal{S}_{n,\delta}$. We show a similar result in our setting, not only for $\mathcal{A} = \hat{w}_{MN}$ but indeed for many interpolation methods.[4]

**Theorem 3.3.** *In Setting **B**, let $\mathcal{A}$ be an algorithm outputting interpolators, $X\mathcal{A}(X, Y) = Y$, with*

$$\mathcal{A}((X_S, X_J), y)_S = \mathcal{A}((X_S, -X_J), y)_S \quad and \quad \lim_{n \to \infty} \lim_{d_J \to \infty} L_{\mathcal{D}}(\mathcal{A}(X, y)) \stackrel{a.s.}{=} \sigma^2. \tag{6}$$

*For any $\delta \in (0, \frac{1}{2})$ and set of typical training examples $\mathcal{S}_{n,\delta}$ satisfying $\Pr(\mathbf{S} \in \mathcal{S}_{n,\delta}) \geq 1 - \delta$, let $\mathcal{W}_{n,\delta}^{\mathcal{A}} = \{\mathcal{A}(X, Y) : (X, Y) \in \mathcal{S}_{n,\delta}\}$ denote the set of typical outputs. Then*

$$\lim_{n \to \infty} \lim_{d_J \to \infty} \sup_{\mathbf{S} \in \mathcal{S}_{n,\delta}} \sup_{w \in \mathcal{W}_{n,\delta}} |L_{\mathcal{D}}(w) - L_{\mathbf{S}}(w)| \stackrel{a.s.}{\geq} 3\sigma^2. \tag{7}$$

*Proof sketch.* For each $\mathbf{S} = (X, Y) \in \mathcal{S}_{n,\delta}$, let $\tilde{\mathbf{S}} = ((X_S, -X_J), Y)$, which has equal density under $\mathcal{D}$, so that $\mathcal{S}_{n,\delta}$ must contain some $(\mathbf{S}, \tilde{\mathbf{S}})$ pairs. Consider $\tilde{w} = \mathcal{A}(\tilde{\mathbf{S}})$: we know that $L_{\mathcal{D}}(\tilde{w}) \stackrel{a.s.}{\to} \sigma^2$ by assumption, but we will show $\lim_{n \to \infty} \lim_{d_J \to \infty} L_{\mathbf{S}}(\tilde{w}) \stackrel{a.s.}{\geq} 4\sigma^2$.

This is easiest to see in the case when $d_S = 0$, so that $y \sim \mathcal{N}(0, \sigma^2)$ is independent of $x$. Then $-X\tilde{w} = Y$, so that $X\tilde{w} = -Y$, and thus $L_{\mathbf{S}}(\tilde{w}) = \frac{1}{n}\|(-Y) - Y\|^2 = \frac{4}{n}\|Y\|^2 \stackrel{a.s.}{=} 4\sigma^2$.

The general case, in Appendix B.3, shows that since $X_S$ is rank $d_S \ll n$, $\tilde{w}_J$ must be large enough to contribute $4\sigma^2 \frac{n - d_S}{n} \to 4\sigma^2$ to the loss. □

From (2), we can see that $\hat{w}_{MN}$ satisfies the symmetry condition in (6). In fact, Proposition B.3 (in Appendix B.3) shows this is also true of many more algorithms, including interpolators which minimize $\|w\|_1$ (basis pursuit) or even $\|w - w^*\|$: any algorithm that picks the interpolator minimizing $f_S(w_S) + f_J(w_J)$, where each function is convex and $f_J(-w) = f_J(w)$.

The attentive reader may have noticed that Theorem 3.3, like Theorem 3.2, applies only to bounds on $|L_{\mathcal{D}}(w) - L_{\mathbf{S}}(w)|$, whereas the general argument as in (3) only needs to bound $L_{\mathcal{D}}(w) - L_{\mathbf{S}}(w)$. Indeed, the proof of Theorem 3.3 exhibits a hypothesis with low generalization error but high *training* error – not a particularly concerning failure mode. Whenever $\mathcal{A}$ is consistent, it is trivially guaranteed that there is a $\mathcal{W}_{n,\delta}$ where (3) holds with $\epsilon_{\mathcal{W}}(n, \delta) \to L_{\mathcal{D}}(w^*)$, and so Nagarajan and Kolter's approach is not meaningful for one-sided bounds.[5] Thus it is not possible to mathematically rule out

$$\epsilon_{n,\delta} \geq \sup_{\mathbf{S} \in \mathcal{S}_{n,\delta}} \sup_{w \in \mathcal{W}_{n,\delta}^{\mathcal{A}}} L_{\mathcal{D}}(w) \geq \sup_{\mathbf{S} \in \mathcal{S}_{n,\delta}} \sup_{w \in \mathcal{W}_{n,\delta}^{\mathcal{A}}} L_{\mathcal{D}}(w) - L_{\mathbf{S}}(w).$$

that one could prove a one-sided bound on $\sup_{w \in \mathcal{W}} L_\mathcal{D}(w) - L_\mathbf{S}(w)$ using a uniform convergence-type technique. (Again, since one-sided uniform convergence is always a consequence of consistency, this question is essentially one of viewpoint: do you first show uniform convergence and then bound consistency through uniform convergence, or do you establish uniform convergence as a consequence of consistency?) In any case, as argued by Nagarajan and Kolter, existing uniform convergence proofs essentially bound $|L_\mathcal{D}(w) - L_\mathbf{S}(w)|$, not $L_\mathcal{D}(w) - L_\mathbf{S}(w)$.

## 4    Uniform convergence for interpolating predictors

In Setting **B**, we now know it is impossible to prove consistency of $\hat{w}_{MN}$ with a bound on $\sup_{w \in \mathcal{W}} |L_\mathcal{D}(w) - L_\mathbf{S}(w)|$ for any fixed choice of $\mathcal{W}$, and it seems quite unlikely that we can do so with bounds on $\sup_{w \in \mathcal{W}} L_\mathcal{D}(w) - L_\mathbf{S}(w)$ either. However, since we are concerned only with zero-training-error predictors, perhaps we should instead look at bounds on

$$\sup_{\|w\| \leq B, \, L_\mathbf{S}(w)=0} L_\mathcal{D}(w) - L_\mathbf{S}(w). \tag{8}$$

Although $L_\mathbf{S}(w)$ is identically 0 in (8), we write it to emphasize that this is still fundamentally a bound on the generalization gap as in (3). When $L_\mathbf{S}(w) = 0$, of course, one-sided and two-sided convergence become the same. Moreover, when $B = \|\hat{w}_{MN}\|$, (8) becomes identically $L_\mathcal{D}(\hat{w}_{MN})$, which we know from Proposition 2.1 is small. Our questions are (a) whether we could have shown this via uniform convergence, and (b) precisely how small $B$ has to be compared to $\|\hat{w}_{MN}\|$ in order to maintain consistency.

The uniform convergence of (8) is a weaker notion than that of Section 3, as the hypothesis set is sample-dependent. But it is still a standard and common form of "uniform convegnce" at the basis of classical learning theory, and is well understood to be necessary for obtaining tight learning guarantees when we expect the training error to be zero. For example, this is the notion used by Valiant [32] to first establish standard (realizable) PAC-learning guarantees, and is the starting point for standard textbooks, as in Section 2.3.1 of Shalev-Shwartz and Ben-David [28], or Theorem 2.1 of Mohri et al. [19] where that book first introduces the term "uniform convergence bound."

A bound on (8) would be implied by bounds with "optimistic rates" [26, 30], which interpolate between a "fast" rate for $L_\mathcal{D}(w) - L_\mathbf{S}(w)$ and a "slow" one depending on $L_\mathbf{S}(w)$. For instance, the result of [30] implies that if $\xi_n$ is a high-probability upper bound on $\max_{1 \leq i \leq n} \|x_i\|^2$, we have uniformly over all $w$ with $\|w\| \leq B$ that

$$L_\mathcal{D}(w) - L_\mathbf{S}(w) \leq \tilde{\mathcal{O}}_P \left( \frac{1}{n} B^2 \xi_n + \sqrt{L_\mathbf{S}(w) \frac{B^2 \xi_n}{n}} \right). \tag{9}$$

But the hidden constants and logarithmic factors in (9) do not meet our needs: to show consistency (as we discuss shortly) we need an asymptotic coefficient of 1 on $B^2 \xi_n / n$, while [30] showed only an upper bound of $200\,000 \log^3(n)$. It seems likely given their extremely indirect proof technique, though, that a much tighter version holds – especially in the special case of bounded-norm linear predictors for square loss. Given Proposition 3.1, it is reasonable to suspect that something like the following may hold fairly generally:

$$\sup_{\|w\| \leq B, \, L_\mathbf{S}(w)=0} L_\mathcal{D}(w) - L_\mathbf{S}(w) \leq \frac{1}{n} B^2 \xi_n + o_P(1), \tag{$\star$}$$

where here $\xi_n$ might refer either to the high-probability upper bound on $\|x\|^2$ or, for sub-Gaussian data, perhaps simply $\mathbb{E}\|x\|^2$. For either choice of $\xi_n$,[6] by taking $B = \|\hat{w}_{MN}\|$ in Setting **B**, applying Proposition 3.1 then gives us (subject to integrability conditions) that for $\lambda_n = \omega(1)$, $\lambda_n = o(n)$,

$$\lim_{d_J \to \infty} \mathbb{E}\, L_\mathcal{D}(\hat{w}_{MN}) = \lim_{d_J \to \infty} \mathbb{E} \left[ \sup_{\|w\| \leq \|\hat{w}_{MN}\|, \, L_\mathbf{S}(w)=0} L_\mathcal{D}(w) - L_\mathbf{S}(w) \right] \leq \sigma^2 + o(1). \tag{10}$$

But ($\star$) would also do more than this: it makes predictions about the generalization error of interpolators with larger-than-minimal norm, not yet known in the literature. In the setting of Proposition 3.1,

($\star$) would imply that

$$\lim_{d_J \to \infty} \mathbb{E}\left[ \sup_{\|w\| \leq \alpha \|\hat{w}_{MN}\|, L_{\mathbf{S}}(w)=0} L_{\mathcal{D}}(w) - L_{\mathbf{S}}(w) \right] \leq \alpha^2 \left[ \sigma^2 + o(1) \right]. \qquad (11)$$

These predictions are important in their own right: outside of linear models, we rarely expect to obtain the interpolator with *exactly* minimal norm.

## 4.1 Uniform convergence of low-norm interpolators in Setting B

The predictions made in (11) in fact hold, with equality.

**Theorem 4.1.** *In Setting **B** with $\lambda_n = o(n)$, fix a sequence $(\alpha_n) \to \alpha$, with each $\alpha_n \geq 1$. Then*

$$\lim_{n \to \infty} \lim_{d_J \to \infty} \mathbb{E}\left[ \sup_{\|w\| \leq \alpha_n \|\hat{w}_{MN}\|, \, L_{\mathbf{S}}(w)=0} L_{\mathcal{D}}(w) - L_{\mathbf{S}}(w) \right] = \alpha^2 L_{\mathcal{D}}(w^*).$$

The proof of Theorem 4.1 is based on bounding (8) directly, although it will take us several steps to get there which we now outline. Along the way, we provide results, especially Proposition 4.3, which are applicable well beyond Setting **B**.

The first tool we will require in our analysis is the best-conceivable interpolator for a given $X$ and $\mathcal{D}$:

**Definition 4.2.** *The* minimal-risk interpolator *[21, Section 3.3] is*

$$\hat{w}_{MR} = \underset{w \, \text{s.t.} \, Xw=Y}{\arg\min} \, L_{\mathcal{D}}(w) = w^* + \Sigma^{-1} X^{\mathsf{T}} (X \Sigma^{-1} X^{\mathsf{T}})^{-1} E. \qquad (12)$$

**Proposition 4.3.** *In Setting **A**, the expected risk of the minimal-risk interpolator is*

$$\mathbb{E} \, L_{\mathcal{D}}(\hat{w}_{MR}) = \frac{p-1}{p-1-n} L_{\mathcal{D}}(w^*).$$

Because $\hat{w}_{MR}$ has perfect knowledge of $\Sigma$, its expected risk turns out to be independent of $\Sigma$. As $p$ increases for fixed $n$ (the second of the double descents), $\mathbb{E} \, L_{\mathcal{D}}(\hat{w}_{MR})$ thus improves monotonically: $\hat{w}_{MR}$ can pick among more interpolators.

We use $\hat{w}_{MR}$ as a constructive tool in our proofs: Theorem 4.5 expands the generalization gap around a fixed predictor in terms of that predictor's risk, and so the minimal-risk predictor is an obvious choice for understanding the gap. Proposition 4.3 also provides lower bounds on interpolation methods: if $p = \mathcal{O}(n)$, then $\hat{w}_{MR}$ is not consistent, and hence no interpolator is. For instance, LASSO is minimax-optimal and consistent for sparse linear regression when $n = \Theta(p)$ [10, 12, 27, 31, 33, 35], but no interpolation method can be. Muthukumar et al. [21, Section 3] discuss this type of result in detail, including for non-Gaussian data; see also [15].

Our next tool measures how much energy in $\Sigma$ is missed by the sample $X$.

**Definition 4.4.** *The* restricted eigenvalue under interpolation *for covariance $\Sigma$ and design $X$ is*

$$\kappa_X(\Sigma) = \sup_{\|w\|=1, \, Xw=0} w^{\mathsf{T}} \Sigma w.$$

We now have the tools to show the following result, which holds even more generally than Setting **A**.

**Theorem 4.5.** *The following results hold deterministically, viewing $L_{\mathcal{D}}(w)$ simply as a quadratic function $L_{\mathcal{D}}(w^*) + \|w - w^*\|_{\Sigma}$, with no distributional assumptions on **S**.*

*(i) It holds that*

$$\sup_{\substack{\|w\| \leq \|\hat{w}_{MR}\| \\ L_{\mathbf{S}}(w)=0}} L_{\mathcal{D}}(w) - L_{\mathbf{S}}(w) = L_{\mathcal{D}}(\hat{w}_{MR}) + \gamma_n \, \kappa_X(\Sigma) \left[ \|\hat{w}_{MR}\|^2 - \|\hat{w}_{MN}\|^2 \right]$$

*where $1 \leq \gamma_n \leq 4$.*

*If the minimal risk interpolator is consistent, $\mathbb{E} \, L_{\mathcal{D}}(\hat{w}_{MR}) - L_{\mathcal{D}}(w^*) \to 0$, then the class of interpolators with norm less than $\|\hat{w}_{MR}\|$ is uniformly consistent if and only if*

$$\mathbb{E} \, \kappa_X(\Sigma) \cdot \left[ \|\hat{w}_{MR}\|^2 - \|\hat{w}_{MN}\|^2 \right] \to 0.$$

*(ii) Fix a sequence $(B_n)$ such that $B_n \geq \|\hat{w}_{MN}\|$ for all $n$. Then*

$$\sup_{\|w\| \leq B_n,\, L_\mathbf{S}(w)=0} L_\mathcal{D}(w) - L_\mathbf{S}(w) = L_\mathcal{D}(\hat{w}_{MN}) + \kappa_X(\Sigma)\left[B_n^2 - \|\hat{w}_{MN}\|^2\right] + R_n$$

*where* $0 \leq R_n \leq 2\sqrt{\left[L_\mathcal{D}(\hat{w}_{MN}) - L_\mathcal{D}(w^*)\right]\kappa_X(\Sigma)\left[B_n^2 - \|\hat{w}_{MN}\|^2\right]}.$

*If* $\mathbb{E}\, L_\mathcal{D}(\hat{w}_{MN}) - L_\mathcal{D}(w^*) \to 0$, *the class of interpolators with norm less than $B_n$ is thus uniformly consistent if and only if*

$$\mathbb{E}\,\kappa_X(\Sigma) \cdot \left[B_n^2 - \|\hat{w}_{MN}\|^2\right] \to 0.$$

The term $\kappa_X(\Sigma)[B^2 - \|\hat{w}_{MN}\|^2]$ appearing in each bound multiplies $\kappa$, essentially "how much" of $\Sigma$ is orthogonal to the data sample, by the amount of excess norm available inside the norm ball. This result makes us expect that $(\star)$ should in fact hold fairly generally with $\xi_n = n\,\kappa_X(\Sigma)$.

Notice also that, of course, $\|\hat{w}_{MN}\| \leq \|\hat{w}_{MR}\|$; thus when $\hat{w}_{MR}$ is consistent (e.g. via Proposition 4.3) and $\mathbb{E}\,\kappa_X(\Sigma)[\|\hat{w}_{MR}\|^2 - \|\hat{w}_{MN}\|^2] \to 0$, then (i) implies $\hat{w}_{MN}$ is consistent as well.

*Proof sketch.* Let $\hat{w}$ be any particular predictor that interpolates the data, and $F \in \mathbb{R}^{p \times (p-n)}$ be the matrix whose columns form an orthonormal basis of the kernel of $X$. Then (8) can be rewritten as

$$\sup_{u \in \mathbb{R}^{p-n}:\|\hat{w}+Fu\|^2 \leq B^2} \|\hat{w} + Fu - w^*\|_\Sigma^2. \tag{13}$$

This is a quadratic program with a single quadratic constraint, which enjoys strong duality even though it is a convex *maximization* [9, Appendix B]. We thus need analyze only the (much simpler) one-dimensional dual problem. For (ii), we take $\hat{w} = \hat{w}_{MN}$ in (13) and obtain the dual as

$$\inf_{\lambda > \|F^\mathsf{T}\Sigma F\|} L_\mathcal{D}(\hat{w}_{MN}) + \|F^\mathsf{T}\Sigma(\hat{w}_{MN} - w^*)\|_{(\lambda I_{p-n} - F^\mathsf{T}\Sigma F)^{-1}}^2 + \lambda\left[B_n^2 - \|\hat{w}_{MN}\|^2\right].$$

Given consistency, we can show that the second term's contribution is negligible, as

$$\|F^\mathsf{T}\Sigma(\hat{w}_{MN} - w^*)\|^2 \leq \|F^\mathsf{T}\Sigma F\| \cdot [L_\mathcal{D}(\hat{w}_{MN}) - L_\mathcal{D}(w^*)],$$

and $(\lambda I_{p-n} - F^\mathsf{T}\Sigma F)^{-1}$ has controlled eigenvalues so that the Mahalanobis norm is similar to the Euclidean norm. Observing that $\kappa_X(\Sigma) = \|F^\mathsf{T}\Sigma F\|$, the conclusion follows by routine calculations.

Case (i) uses a similar strategy, taking $\hat{w} = \hat{w}_{MR}$. The full proof is given in Appendix C.2. $\qquad\square$

Now, all that remains is to evaluate the relevant quantities in Setting **B**.

**Proposition 4.6.** *In Setting **B** with $\lambda_n = o(n)$,*

$$\lim_{n \to \infty} \lim_{d_J \to \infty} \mathbb{E}\left[\sup_{\|w\| \leq \|\hat{w}_{MR}\|,\, L_\mathbf{S}(w)=0} L_\mathcal{D}(w) - L_\mathbf{S}(w)\right] = L_\mathcal{D}(w^*).$$

*Proof sketch for Theorem 4.1 and Proposition 4.6.* We apply Theorem 4.5. With probability one,

$$\lim_{d_J \to \infty} \kappa_X(\Sigma) = \frac{\lambda_n}{n} \left\|\left[\frac{X_S^\mathsf{T} X_S}{n} + \frac{\lambda_n}{n} I_{d_S}\right]^{-1}\right\|.$$

As the first term inside the inverse converges to $I_{d_S}$ and the second term vanishes, we can expect $\kappa_X(\Sigma) \approx \lambda_n/n$. We bound the other terms by observing that there exists a sequence $\beta_n \to 1$ with

$$\lim_{d_J \to \infty} \mathbb{E}\|\hat{w}_{MR}\|^2 = \|w_S^*\|^2 + \frac{\sigma^2 n}{\lambda_n}$$

$$\lim_{d_J \to \infty} \mathbb{E}\|\hat{w}_{MN}\|^2 = \|w^*\|^2 + \sigma^2 \frac{n - d_S}{\lambda_n} + \beta_n\left(\frac{\sigma^2 d_S - \lambda_n \|w_S^*\|^2}{n}\right),$$

so $\lim_{d_J \to \infty} \mathbb{E}\left[\|\hat{w}_{MR}\|^2 - \mathbb{E}\|\hat{w}_{MN}\|^2\right] = \sigma^2 d_S/\lambda_n + \mathcal{O}\left(\lambda_n \|w^*\|^2/n\right).$

Because $\hat{w}_{MR}$ is consistent via Proposition 4.3, this proves Proposition 4.6. As $\|\hat{w}_{MN}\| \leq \|\hat{w}_{MR}\|$, this further implies $\hat{w}_{MN}$ is consistent, so that the $R_n$ term of Theorem 4.5 (ii) vanishes. $\qquad\square$

We can see that $\kappa_X(\Sigma)$ tends to 0 while $\|\hat{w}_{MN}\|$ explodes, and in Setting **B** their product turns out to converge to *exactly* the Bayes risk. Because the other terms of Theorem 4.5 (ii) cancel, this gives us precisely the tight result we need for Theorem 4.1, and further suggests that the speculative upper bound $\kappa_X(\Sigma)B^2$ probably holds in more general settings.

We have at last shown in Theorem 4.1 a uniform convergence bound not only showing consistency of $\hat{w}_{MN}$, but furthermore verifying the predictions of (11). Thus if we obtain an interpolator with norm $1.1\|\hat{w}_{MN}\|$, we will suffer at most $1.21\sigma^2$ asymptotic risk. If we obtain an interpolator with norm no more than a constant amount larger than the minimal norm, we achieve asymptotic consistency.

## 5   Discussion

In this work, we shed new light on uniform convergence and its relationship to interpolation learning. We show that uniform control of the generalization gap cannot explain interpolation learning, for almost *any* interpolator, even in a simple setting. But we argue that when discussing "uniform convergence" in the context of interpolation learning, we should slightly broaden our horizons to include interpolation-specific uniform convergence bounds such as ($\star$), or more generally "optimistic" (training-error-dependent) bounds [26, 30]. We show that despite recent sentiments to the contrary, such bounds *could* in principal explain interpolation learning, by demonstrating this in the "junk features" setting. Doing so requires obtaining very tight bounds, include tight constants – perhaps a difficult task, but not impossible. (For example, for linear predictors with a Lipschitz loss in a non-realizable setting, we do know the exact worst-case bound, with a tight numeric constant [16].)

Our results are also of independent interest in ensuring success with interpolation learning: in settings other than linear regression, where a closed-form solution is available, it is generally unlikely in practice that we find the *exact* minimum-norm solution. (Even gradient descent for linear regression would find this only when initialized exactly in the span of the data; other forms of implicit bias are likewise suboptimal.) Our results give some reassurance that, at least in this simple setting, approximately minimizing the norm is sufficient. The natural next step in this vein would be to study predictors with small but nonzero loss. This could either be done directly in the style of our Theorem 4.1, or by providing an optimistic rate as in (9) with tight constants. Our specific techniques, as well as the general takeaway of considering interpolation-specific bounds, could also be potentially applicable to settings beyond linear regression, especially the idea of studying the generalization gap via the dual problem: although strong duality may not be available in more general settings, upper bounds are always possible with weak duality.

### Broader Impact

Interpolation learning is currently thought to be one of the core mysteries standing between us and a theoretical understanding of modern deep learning. Although there has recently been some key progress, many challenges remain. Our paper, in advancing the study of interpolation learning, makes another step on the path towards understanding the deep learning models that are quickly becoming ubiquitous throughout society, whether we understand them or not. In our view, increased understanding of these models can lead to safer, more reliable, and more controlled deployment, especially in sensitive domains.

In particular, we discuss a key component of statistical learning theory, namely uniform convergence, whose relevance to deep learning in general – and interpolation learning specifically – has recently been questioned. We make an explicit connection between the work on interpolation learning and the recent notion of "algorithmic dependent uniform convergence" [22]. Instead of outright dismissal, we show that a more nuanced view is appropriate. By doing so, we hope to help guide the re-pivoting that statistical learning theory is currently undergoing.

We emphasize that, despite providing some positive theoretical results, we are certainly not advocating for preferring interpolation methods over other approaches. In particular, the increased sensitivity of interpolation methods may have problematic ramifications for robustness or privacy.

## Acknowledgments and Disclosure of Funding

Research supported in part by NSF IIS award 1764032 and NSF HDR TRIPODS award 1934843.

## Footnotes

[1]Talk at the Simons Institute for the Theory of Computing, July 2019: simons.berkeley.edu/talks/tbd-65

[2]If the noise scaling were $\omega(1/d_J)$, then as $d_J \to \infty$, the minimal-norm solution would exploit the exploding magnitude of the noise components, and all of the signal would "bleed" into the noise dimensions [14, 21], giving $\|\hat{w}_{MN}\| \to 0$ and $L_{\mathcal{D}}(\hat{w}_{MN}) \to L_{\mathcal{D}}(0_p)$ – in the ridge regression equivalence, we let the regularization weight go to infinity. On the other hand, if the noise scaling were $o(1/d_J)$, then we would have $\|\hat{w}_{MN}\| \to \infty$, significantly complicating matters. $\Theta(1/d_J)$ is the only scaling in which $\|\hat{w}_{MN}\|$ is bounded but nonzero.

[3]Both exchanges can be justified using dominated convergence thoerem and the techniques from the proof of Proposition 4.6, which independently shows a stronger statement.

[4]Lemma 5.2 of Negrea et al. [24] is closely related; it covers Setting **A** in general, but applies only to $\hat{w}_{MN}$ and shows a smaller gap.

[5]Take $\mathcal{S}_{n,\delta} = \{(X, Y) : L_{\mathcal{D}}(X, Y) \leq L_{\mathcal{D}}(w^*) + \epsilon_{n,\delta}\}$; consistency implies that there is a choice of $\epsilon_{n,\delta} \to 0$ such that $\Pr(\mathbf{S} \in \mathcal{S}_{n,\delta}) \geq 1 - \delta$ and

[6]If $\xi_n$ is a high-probability upper bound, we further require $\lambda_n = \omega(\log n)$.

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
