[Supplementary Material]

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

[7]We use the following version of the theorem, which is slightly more general than the usual one. Suppose there exists a sequence of $l_1$ random variables $Y_n$ such that $Y_n \geq X_n$ and

[8]Using standard properties of the inverse Wishart distribution, we can check that

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

## A Proofs for Section 2

**Lemma 2.3.** *In Setting B, if $\lambda_n = o(n)$, then $\lim_{n\to\infty} \mathbb{E}\left[L_\mathcal{D}(\hat{w}_{\lambda_n}) - L_\mathcal{D}(w^*)\right] = 0$.*

*Proof.* We can write

$$
\begin{aligned}
\hat{w}_{\lambda_n} - w_S^* &= (X_S^\mathsf{T} X_S + \lambda_n I_{d_S})^{-1} X_S^\mathsf{T}(X_S w_S^* + E) - w_S^* \\
&= ((X_S^\mathsf{T} X_S + \lambda_n I_{d_S})^{-1} X_S^\mathsf{T} X_S - I_{d_S}) w_S^* + (X_S^\mathsf{T} X_S + \lambda_n I_{d_S})^{-1} X_S^\mathsf{T} E \\
&= \left[\left(\frac{X_S^\mathsf{T} X_S}{n} + \frac{\lambda_n}{n} I_{d_S}\right)^{-1} \frac{X_S^\mathsf{T} X_S}{n} - I_{d_S}\right] w_S^* + \left(\frac{X_S^\mathsf{T} X_S}{n} + \frac{\lambda_n}{n} I_{d_S}\right)^{-1} \frac{X_S^\mathsf{T} E}{n}.
\end{aligned}
$$

Therefore, by independence of $X_S$ and $E$,

$$
\begin{aligned}
\mathbb{E}[L_\mathcal{D}(\hat{w}_{\lambda_n}) - L_\mathcal{D}(w^*)] &= \mathbb{E}\|\hat{w}_{\lambda_n} - w_S^*\|^2 \\
&= \mathbb{E}\left\|\left[\left(\frac{X_S^\mathsf{T} X_S}{n} + \frac{\lambda_n}{n} I_{d_S}\right)^{-1} \frac{X_S^\mathsf{T} X_S}{n} - I_{d_S}\right] w_S^*\right\|^2 + \mathbb{E}\left\|\left(\frac{X_S^\mathsf{T} X_S}{n} + \frac{\lambda_n}{n} I_{d_S}\right)^{-1} \frac{X_S^\mathsf{T} E}{n}\right\|^2 \\
&= \mathbb{E}\left\|\left[\left(\frac{X_S^\mathsf{T} X_S}{n} + \frac{\lambda_n}{n} I_{d_S}\right)^{-1} \frac{X_S^\mathsf{T} X_S}{n} - I_{d_S}\right] w_S^*\right\|^2 + \sigma^2 \mathbb{E}\frac{1}{n}\operatorname{Tr}\left[\left(\frac{X_S^\mathsf{T} X_S}{n} + \frac{\lambda_n}{n} I_{d_S}\right)^{-2} \frac{X_S^\mathsf{T} X_S}{n}\right].
\end{aligned}
$$

Write the SVD for $X_S = UDV^\mathsf{T}$. Since $X_S$ has rank at most $d_S$, we denote its singular values as $\sqrt{\rho_1}, ..., \sqrt{\rho_{d_S}}$, and

$$
\|(X_S^\mathsf{T} X_S + \lambda I_{d_S})^{-1} X_S^\mathsf{T} X_S\| = \|(D^\mathsf{T} D + \lambda I_{d_S})^{-1} D^\mathsf{T} D\| = \max_{i\in[p]} \frac{\rho_i}{\lambda_n + \rho_i} \le 1.
$$

Thus, we have

$$
\left\|\left[\left(\frac{X_S^\mathsf{T} X_S}{n} + \frac{\lambda}{n} I_{d_S}\right)^{-1} \frac{X_S^\mathsf{T} X_S}{n} - I_{d_S}\right] w_S^*\right\|^2 \le (1+1)^2 \|w_S^*\|^2 = 4\|w_S^*\|^2
$$

which is clearly integrable.

As $d_S$ stays fixed as $n \to \infty$, by the strong law of large numbers we have $\frac{X_S^\mathsf{T} X_S}{n} \to I_{d_S}$. Assuming that $\frac{\lambda_n}{n} \to \gamma$, then by the continuous mapping and dominated convergence theorems, the first term converges to

$$
\mathbb{E}\lim_{n\to\infty} \left\|\left[1 - (1+\gamma)^{-1}\right] w_S^*\right\|^2 = \left(\frac{\gamma}{1+\gamma} \cdot \|w_S^*\|\right)^2,
$$

Moreover, it holds that

$$
\begin{aligned}
\frac{1}{n}\operatorname{Tr}\left[\left(\frac{X_S^\mathsf{T} X_S}{n} + \frac{\lambda_n}{n} I_{d_S}\right)^{-2} \frac{X_S^\mathsf{T} X_S}{n}\right] &= \sum_{i=1}^{d_S}\left(\frac{\sqrt{\rho_i}}{\rho_i + \lambda_n}\right)^2 \\
&\le \sum_{i=1}^{d_S}\frac{1}{\rho_i} = \operatorname{Tr}\left[(X_S^\mathsf{T} X_S)^{-1}\right]
\end{aligned}
$$

Using the first moment of inverse Wishart distribution, the second term can be controlled by

$$
\sigma^2 \mathbb{E}\operatorname{Tr}\left[(X_S^\mathsf{T} X_S)^{-1}\right] = \sigma^2 \frac{d_S}{n - d_S - 1} \to 0
$$

Note that the first term converges to 0 as long as $\gamma = 0$, and the desired conclusion follows.  □

# B   Proofs for Section 3

## B.1   Size of the minimal-norm interpolator (Proposition 3.1)

**Proposition B.1.** *In Setting B, it holds that*

$$\lim_{d_J \to \infty} \mathbb{E}\|\hat{w}_{MR}\|^2 = \|w^*\|^2 + \frac{\sigma^2 n}{\lambda_n}.$$

*Moreover, there exists a sequence $(\beta_n)$ such that $\beta_n \to 1$ and*

$$\lim_{d_J \to \infty} \mathbb{E}\|\hat{w}_{MN}\|^2 = \|w^*\|^2 + \sigma^2 \frac{n - d_S}{\lambda_n} + \beta_n \left( \frac{\sigma^2 d_S - \lambda_n \|w_S^*\|^2}{n} \right).$$

*Consequently, we have*

$$\lim_{d_J \to \infty} \mathbb{E}\left[ \|\hat{w}_{MR}\|^2 - \|\hat{w}_{MN}\|^2 \right] = \frac{\sigma^2 d_S}{\lambda_n} + \beta_n \left( \frac{\lambda_n \|w_S^*\|^2 - \sigma^2 d_S}{n} \right).$$

*Proof.* Let $\{e_i\}$ be the standard basis in $\mathbb{R}^p$ and write $\Sigma = \sum_{i=1}^p \mu_i e_i e_i^T$, with $\mu_i = 1$ for $1 \le i \le d_S$ and $\mu_i = \lambda_n / d_J$ for $i > d_S$. By independence of $X$ and $E$, we have

$$\begin{aligned}
\mathbb{E}\|\hat{w}_{MR}\|^2 &= \|w^*\|^2 + \mathbb{E}\|\Sigma^{-1} X^{\mathsf{T}} (X \Sigma^{-1} X^{\mathsf{T}})^{-1} E\|^2 \\
&= \|w^*\|^2 + \sigma^2 \mathbb{E}\left[ \mathrm{Tr}\left( (ZZ^{\mathsf{T}})^{-1} (Z \Sigma^{-1} Z^{\mathsf{T}}) (ZZ^{\mathsf{T}})^{-1} \right) \right] \\
&= \|w^*\|^2 + \sum_{i=1}^p \frac{\sigma^2}{\mu_i} \mathbb{E}\left[ \|(ZZ^T)^{-1} Z e_i\|^2 \right].
\end{aligned}$$

By rotational invariance of the standard normal distribution for $Z$, we have

$$\mathbb{E}\left[ \|(ZZ^T)^{-1} Z e_i\|^2 \right] = \frac{\mathbb{E}\,\mathrm{Tr}(Z^T (ZZ^T)^{-2} Z)}{p} = \frac{\mathbb{E}\,\mathrm{Tr}((ZZ^T)^{-1})}{p} = \frac{n}{p(p - n - 1)}.$$

Plugging in, we get

$$\begin{aligned}
\mathbb{E}\|\hat{w}_{MR}\|^2 &= \|w^*\|^2 + \left( \sum_{i=1}^p \frac{\sigma^2}{\mu_i} \right) \frac{n}{p(p - n - 1)} \\
&= \|w^*\|^2 + \sigma^2 \left( d_S + \frac{d_J^2}{\lambda_n} \right) \frac{n}{p(p - n - 1)}.
\end{aligned}$$

Sending $d_J \to \infty$ and recalling $p = d_S + d_J$, we obtain

$$\lim_{d_J \to \infty} \mathbb{E}\|\hat{w}_{MR}\|^2 = \|w^*\|^2 + \frac{\sigma^2 n}{\lambda_n}.$$

Moreover, it holds that

$$\begin{aligned}
\|\hat{w}_{MR}\|^2 &= \|w^*\|^2 + \mathrm{Tr}\left( (ZZ^{\mathsf{T}})^{-1} (Z \Sigma^{-1} Z^{\mathsf{T}}) (ZZ^{\mathsf{T}})^{-1} E E^{\mathsf{T}} \right) + 2\langle w^*, \Sigma^{-1/2} Z^{\mathsf{T}} (ZZ^{\mathsf{T}})^{-1} E \rangle \\
&= \|w^*\|^2 + \mathrm{Tr}\left( \left( \frac{ZZ^{\mathsf{T}}}{p} \right)^{-1} \left( \frac{Z \Sigma^{-1} Z^{\mathsf{T}}}{p^2} \right) \left( \frac{ZZ^{\mathsf{T}}}{p} \right)^{-1} E E^{\mathsf{T}} \right) + 2\left\langle \frac{Z \Sigma^{-1/2} w^* E^{\mathsf{T}}}{p}, \left( \frac{ZZ^{\mathsf{T}}}{p} \right)^{-1} \right\rangle.
\end{aligned}$$

Notice that

$$\lim_{d_J \to \infty} \left( \frac{ZZ^{\mathsf{T}}}{p} \right)^{-1} \overset{a.s.}{=} I_n$$

$$\lim_{d_J \to \infty} \frac{Z \Sigma^{-1} Z^{\mathsf{T}}}{p^2} = \lim_{d_J \to \infty} \frac{1}{p^2} \left( Z_S Z_S^{\mathsf{T}} + \frac{d_J^2}{\lambda_n} \frac{Z_J Z_J^{\mathsf{T}}}{d_J} \right) \overset{a.s.}{=} \frac{1}{\lambda_n} I_n$$

$$Z \Sigma^{-1/2} w^* E^{\mathsf{T}} = \begin{bmatrix} Z_S & Z_J \end{bmatrix} \begin{bmatrix} I_{d_S} & 0_{d_S \times d_J} \\ 0_{d_J \times d_S} & \sqrt{\frac{d_J}{\lambda_n}} I_{d_J} \end{bmatrix} \begin{bmatrix} w_S^* \\ 0_{d_J} \end{bmatrix} E^{\mathsf{T}} = Z_S w_S^* E^{\mathsf{T}} \implies \frac{Z \Sigma^{-1/2} w^* E^{\mathsf{T}}}{p} \overset{a.s.}{=} 0.$$

Plugging in, we obtain

$$\lim_{d_J \to \infty} \|\hat{w}_{MR}\|^2 \overset{a.s.}{=} \|w^*\|^2 + \frac{\|E\|^2}{\lambda_n}, \quad \text{and so} \quad \mathbb{E}\left[\lim_{d_J \to \infty} \|\hat{w}_{MR}\|^2\right] = \lim_{d_J \to \infty} \mathbb{E}\|\hat{w}_{MR}\|^2.$$

Clearly, the sequence of random variables $(\|\hat{w}_{MR}\|^2)$ as we let $d_J \to \infty$ dominates $(\|\hat{w}_{MN}\|^2)$. By the dominated convergence theorem [7]

$$\begin{aligned}
\lim_{d_J \to \infty} \mathbb{E}\|\hat{w}_{MN}\|^2 &= \mathbb{E}\left[\lim_{d_J \to \infty} \|\hat{w}_{MN}\|^2\right] \\
&= \mathbb{E}\left[\lim_{d_J \to \infty} (X_S w_S^* + E)^\mathsf{T}(XX^\mathsf{T})^{-1}XX^\mathsf{T}(XX^\mathsf{T})^{-1}(X_S w_S^* + E)\right] \\
&= \mathbb{E}\left[\lim_{d_J \to \infty} (X_S w_S^* + E)^\mathsf{T}(X_S X_S^\mathsf{T} + X_J X_J^\mathsf{T})^{-1}(X_S w_S^* + E)\right] \\
&= \mathbb{E}\left[(X_S w_S^* + E)^\mathsf{T}(X_S X_S^\mathsf{T} + \lambda_n I_n)^{-1}(X_S w_S^* + E)\right] \\
&= (w_S^*)^\mathsf{T}\mathbb{E}[X_S^\mathsf{T}(X_S X_S^\mathsf{T} + \lambda_n I_n)^{-1}X_S]w_S^* + \sigma^2\, \mathbb{E}\operatorname{Tr}\left((X_S X_S^\mathsf{T} + \lambda_n I_n)^{-1}\right).
\end{aligned}$$

With probability one, $X_S X_S^\mathsf{T}$ is a $n \times n$ matrix with rank $d_S$, so the eigenvalues of $(X_S X_S^\mathsf{T} + \lambda_n I_n)^{-1}$ consist of the $d_S$ eigenvalues of $(X_S^\mathsf{T} X_S + \lambda_n I_{d_S})^{-1}$ and $(n - d_S)$ copies of $\frac{1}{0+\lambda_n}$. This implies

$$\sigma^2\, \mathbb{E}\operatorname{Tr}\left((X_S X_S^\mathsf{T} + \lambda I_n)^{-1}\right) = \sigma^2\, \mathbb{E}\operatorname{Tr}\left((X_S^\mathsf{T} X_S + \lambda I_{d_S})^{-1}\right) + \sigma^2 \frac{n - d_S}{\lambda_n}.$$

Moreover, by the rotational invariance of $X_S \sim \mathcal{N}(0, I_{d_S})$,

$$\begin{aligned}
(w_S^*)^\mathsf{T}\mathbb{E}[X_S^\mathsf{T}(X_S X_S^\mathsf{T} + \lambda_n I_n)^{-1}X_S]w_S^* &= \frac{\|w_S^*\|^2}{d_S}\, \mathbb{E}\operatorname{Tr}\left(X_S^\mathsf{T}(X_S X_S^\mathsf{T} + \lambda_n I_n)^{-1}X_S\right) \\
&= \frac{\|w_S^*\|^2}{d_S}\, \mathbb{E}\operatorname{Tr}\left(X_S^\mathsf{T} X_S(X_S^\mathsf{T} X_S + \lambda_n I_{d_S})^{-1}\right) \\
&= \frac{\|w_S^*\|^2}{d_S}\, \mathbb{E}\operatorname{Tr}\left(I_{d_S} - \lambda_n(X_S^\mathsf{T} X_S + \lambda_n I_{d_S})^{-1}\right) \\
&= \|w_S^*\|^2 - \frac{\lambda_n\|w_S^*\|^2}{d_S}\, \mathbb{E}\operatorname{Tr}\left((X_S^\mathsf{T} X_S + \lambda_n I_{d_S})^{-1}\right).
\end{aligned}$$

Plugging in, we get

$$\begin{aligned}
\lim_{d_J \to \infty} \mathbb{E}\|\hat{w}_{MN}\|^2 &= \|w^*\|^2 + \sigma^2 \frac{n - d_S}{\lambda_n} + \left(\sigma^2 - \frac{\lambda_n\|w_S^*\|^2}{d_S}\right)\mathbb{E}\operatorname{Tr}\left((X_S^\mathsf{T} X_S + \lambda_n I_{d_S})^{-1}\right) \\
&= \|w^*\|^2 + \sigma^2 \frac{n - d_S}{\lambda_n} + \left(\frac{\sigma^2 d_S - \lambda_n\|w_S^*\|^2}{n}\right) \cdot \left[\frac{\mathbb{E}\operatorname{Tr}\left(\left(\frac{X_S^\mathsf{T} X_S}{n} + \frac{\lambda_n}{n} I_{d_S}\right)^{-1}\right)}{d_S}\right].
\end{aligned}$$

$$\lim_{n \to \infty} \mathbb{E}\, Y_n = \mathbb{E}\lim_{n \to \infty} Y_n;$$

then we have

$$\lim_{n \to \infty} \mathbb{E}\, X_n = \mathbb{E}\lim_{n \to \infty} X_n.$$

The proof is essentially the same and applies Fatou's lemma to $X_n$ and $Y_n - X_n$.

As $\text{Tr}\left(\left(\frac{X_S^\mathsf{T} X_S}{n}\right)^{-1}\right)$, which has limit $d_S$ in expectation,[8] dominates $\text{Tr}\left(\left(\frac{X_S^\mathsf{T} X_S}{n} + \frac{\lambda_n}{n} I_{d_S}\right)^{-1}\right)$, by the dominated convergence theorem

$$\lim_{n\to\infty} \frac{1}{d_S} \mathbb{E}\,\text{Tr}\left(\left(\frac{X_S^\mathsf{T} X_S}{n} + \frac{\lambda_n}{n} I_{d_S}\right)^{-1}\right) = 1.$$

Letting the term in brackets be $\beta_n$, we have the result. $\qquad\square$

**Proposition 3.1.** *As $n \to \infty$ in Setting **B**, if $\lambda_n$ is both $o(n)$ and $\omega(1)$, then*

$$\lim_{d_J \to \infty} \mathbb{E}\|\hat{w}_{MN}\|^2 = \frac{\sigma^2 n}{\lambda_n} + \mathcal{O}(1) \quad and \quad \lim_{d_J \to \infty} \frac{(\mathbb{E}\|\hat{w}_{MN}\|^2)(\mathbb{E}\|x\|^2)}{n} = \sigma^2 + o(1).$$

*Proof.* By Proposition B.1, there exists a sequence $(\beta_n)$ such that $\beta_n \to 1$ and

$$\lim_{d_J \to \infty} \mathbb{E}\|\hat{w}_{MN}\|^2 = \sigma^2 \frac{n}{\lambda_n} + \left[\|w^*\|^2 - \sigma^2 \frac{d_S}{\lambda_n} + \beta_n \left(\frac{\sigma^2 d_S - \lambda_n \|w_S^*\|^2}{n}\right)\right].$$

Moreover, we have

$$\mathbb{E}\|x\|^2 = \text{Tr}(\Sigma) = d_S \cdot 1 + d_J \cdot \frac{\lambda_n}{d_J} = d_S + \lambda_n.$$

Plugging in, we obtain

$$\frac{(\mathbb{E}\|\hat{w}_{MN}\|^2)(\mathbb{E}\|x\|^2)}{n} = \sigma^2 \frac{d_S + \lambda_n}{\lambda_n} + \frac{d_S + \lambda_n}{n}\left[\|w^*\|^2 - \sigma^2 \frac{d_S}{\lambda_n} + \beta_n \left(\frac{\sigma^2 d_S - \lambda_n \|w_S^*\|^2}{n}\right)\right].$$

By assumption, $1/\lambda_n \to 0$ and $\lambda_n/n \to 0$; thus the dominant term inside the brackets is $\|w^*\|^2 = \mathcal{O}(1)$. The conclusion follows by

$$\frac{d_S + \lambda_n}{\lambda_n} \to 1 \quad \text{and} \quad \frac{d_S + \lambda_n}{n} \to 0. \qquad\square$$

## B.2 Divergence of the generalization gap of norm balls (Section 3.1)

**Proposition B.2.** *Let $\rho(\Sigma - \hat{\Sigma})$ be the algebraically largest eigenvalue of $\Sigma - \hat{\Sigma}$. It holds that*

$$\sup_{\|w\| \le \|\hat{w}_{MN}\|} L_{\mathcal{D}}(w) - L_{\mathbf{S}}(w) \ge \rho(\Sigma - \hat{\Sigma}) \cdot (\|\hat{w}_{MN}\| - \|w^*\|)^2 + \left[L_{\mathcal{D}}(w^*) - \frac{1}{n}\|E\|^2\right]$$

*and similarly for two sided uniform convergence, it holds that*

$$\sup_{\|w\| \le \|\hat{w}_{MN}\|} |L_{\mathcal{D}}(w) - L_{\mathbf{S}}(w)| \ge \|\Sigma - \hat{\Sigma}\| \cdot (\|\hat{w}_{MN}\| - \|w^*\|)^2 - \left|L_{\mathcal{D}}(w^*) - \frac{\|E\|^2}{n}\right|.$$

*Proof.* Recall from (1) that

$$\begin{aligned}
L_{\mathbf{S}}(w) &= \frac{1}{n}\|Xw - Y\|^2 \\
&= \frac{1}{n}\|X(w - w^*) + Xw^* - Y\|^2 \\
&= (w - w^*)^\mathsf{T} \hat{\Sigma}(w - w^*) + \frac{\|E\|^2}{n} - 2\left\langle w - w^*, \frac{X^\mathsf{T} E}{n}\right\rangle.
\end{aligned}$$

$$\lim_{n\to\infty} \mathbb{E}\,\text{Tr}\left(\left(\frac{X_S^\mathsf{T} X_S}{n}\right)^{-1}\right) = d_S = \mathbb{E}\lim_{n\to\infty}\text{Tr}\left(\left(\frac{X_S^\mathsf{T} X_S}{n}\right)^{-1}\right).$$

Therefore, we can decompose the generalization gap as

$$
\begin{aligned}
L_{\mathcal{D}}(w) - L_{\mathbf{S}}(w) &= L_{\mathcal{D}}(w^*) + (w - w^*)^\mathsf{T}\Sigma(w - w^*) - L_{\mathbf{S}}(w) \\
&= \left[ L_{\mathcal{D}}(w^*) - \frac{\|E\|^2}{n} \right] + (w - w^*)^\mathsf{T}(\Sigma - \hat{\Sigma})(w - w^*) + 2\left\langle w - w^*, \frac{X^\mathsf{T}E}{n} \right\rangle.
\end{aligned}
$$

Observe that

$$
\begin{aligned}
\sup_{\|w\| \le \|\hat{w}_{MN}\|} (w - w^*)^\mathsf{T}(\Sigma - \hat{\Sigma})(w - w^*) &+ 2\left\langle w - w^*, \frac{X^\mathsf{T}E}{n} \right\rangle \\
&\ge \sup_{\|w\| \le \|\hat{w}_{MN}\| - \|w^*\|} w^\mathsf{T}(\Sigma - \hat{\Sigma})w + 2\left\langle w, \frac{X^\mathsf{T}E}{n} \right\rangle \\
&\ge \rho(\Sigma - \hat{\Sigma}) \cdot (\|\hat{w}_{MN}\| - \|w^*\|)^2.
\end{aligned}
$$

The last inequality holds by picking $w$ to be $\pm(\|\hat{w}_{MN}\| - \|w^*\|)$ times the top eigenvector of $\Sigma - \hat{\Sigma}$ for whichever sign makes the linear term nonnegative. By the same reasoning, we have

$$
\sup_{\|w\| \le \|\hat{w}_{MN}\|} |L_{\mathcal{D}}(w) - L_{\mathbf{S}}(w)| \ge \|\Sigma - \hat{\Sigma}\| \cdot (\|\hat{w}_{MN}\| - \|w^*\|)^2 - \left| L_{\mathcal{D}}(w^*) - \frac{\|E\|^2}{n} \right|. \qquad \square
$$

**Theorem 3.2.** *In Setting **B**, if $\lambda_n = o(n)$ then*

$$
\lim_{n \to \infty} \lim_{d_J \to \infty} \mathbb{E}\left[ \sup_{\|w\| \le \|\hat{w}_{MN}\|} |L_{\mathcal{D}}(w) - L_{\mathbf{S}}(w)| \right] = \infty.
$$

*Proof.* We will show that in Setting **B** as long as $\lambda_n = o(n)$,

$$
\lim_{n \to \infty} \lim_{d_J \to \infty} \mathbb{E}\|\Sigma - \hat{\Sigma}\| \cdot \|\hat{w}_{MN}\|^2 = \infty.
$$

By Fatou's lemma and the calculation in Proposition B.1,

$$
\begin{aligned}
\lim_{d_J \to \infty} \mathbb{E}\|\Sigma - \hat{\Sigma}\| \cdot \|\hat{w}_{MN}\|^2 &\ge \mathbb{E} \lim_{d_J \to \infty} \|\Sigma - \hat{\Sigma}\| \cdot \|\hat{w}_{MN}\|^2 \\
&= \mathbb{E} \lim_{d_J \to \infty} \|\Sigma - \hat{\Sigma}\| \cdot \left( (X_S w_S^* + E)^\mathsf{T}(X_S X_S^\mathsf{T} + \lambda_n I_n)^{-1}(X_S w_S^* + E) \right).
\end{aligned}
$$

By independence of $X$ and $E$, we have

$$
\begin{aligned}
\lim_{d_J \to \infty} \mathbb{E}\left[ \|\Sigma - \hat{\Sigma}\| \cdot \|\hat{w}_{MN}\|^2 \right] &\ge \mathbb{E} \lim_{d_J \to \infty} \|\Sigma - \hat{\Sigma}\| \cdot \left( E^\mathsf{T}(X_S X_S^\mathsf{T} + \lambda_n I_n)^{-1} E \right) \\
&= \sigma^2 \mathbb{E}\left[ \lim_{d_J \to \infty} \|\Sigma - \hat{\Sigma}\| \cdot \mathrm{Tr}\left( (X_S X_S^\mathsf{T} + \lambda_n I_n)^{-1} \right) \right] \\
&\ge \sigma^2 \mathbb{E}\left[ \lim_{d_J \to \infty} \|\Sigma - \hat{\Sigma}\| \cdot \left( \frac{n - d_S}{\lambda_n} \right) \right] \\
&= \left( \sigma^2 \frac{n - d_S}{\lambda_n} \right) \mathbb{E}\left[ \lim_{d_J \to \infty} \|\Sigma - \hat{\Sigma}\| \right].
\end{aligned}
$$

Next we want to interchange limit and expectation. Note that

$$
\begin{aligned}
\|\Sigma - \hat{\Sigma}\| &\le \|\Sigma\| + \|\hat{\Sigma}\| \\
&= \|\Sigma\| + \left\| \frac{X_S^\mathsf{T}X_S + X_J X_J^\mathsf{T}}{n} \right\| \\
&\le \|\Sigma\| + \left\| \frac{X_S^\mathsf{T}X_S}{n} \right\| + \mathrm{Tr}\left( \frac{X_J X_J^\mathsf{T}}{n} \right) \\
&= \|\Sigma\| + \left\| \frac{X_S^\mathsf{T}X_S}{n} \right\| + \frac{\lambda_n}{n} \mathrm{Tr}\left( \frac{Z_J Z_J^\mathsf{T}}{d_J} \right).
\end{aligned}
$$

The first two terms do not depend on $d_J$. It is easy to verify that

$$\lim_{d_J \to \infty} \mathbb{E}\left[\frac{\lambda_n}{n}\operatorname{Tr}\left(\frac{Z_J Z_J^\mathsf{T}}{d_J}\right)\right] = \lambda_n = \mathbb{E}\left[\lim_{d_J \to \infty}\frac{\lambda_n}{n}\operatorname{Tr}\left(\frac{Z_J Z_J^\mathsf{T}}{d_J}\right)\right]$$

as $\frac{Z_J Z_J^\mathsf{T}}{d_J} \overset{a.s.}{\to} I_n$. Therefore, by the dominated convergence theorem

$$\lim_{d_J \to \infty} \mathbb{E}\left[\|\Sigma - \hat{\Sigma}\| \cdot \|\hat{w}_{MN}\|^2\right] \geq \lim_{d_J \to \infty}\left(\sigma^2 \frac{n - d_S}{\lambda_n}\right)\mathbb{E}\|\Sigma - \hat{\Sigma}\|.$$

Koltchinskii and Lounici [17] show that, for Gaussian data,

$$\mathbb{E}\|\Sigma - \hat{\Sigma}\| \geq C \max\left(\sqrt{\frac{\operatorname{Tr}(\Sigma)\,\|\Sigma\|}{n}}, \frac{\operatorname{Tr}(\Sigma)}{n}\right),$$

where $C$ is a universal constant. Thus, in our case

$$\mathbb{E}\|\Sigma - \hat{\Sigma}\| \geq C\sqrt{\frac{d_S + \lambda_n}{n}}.$$

Since $\lambda_n = o(n)$, this implies

$$\lim_{n \to \infty}\lim_{d_J \to \infty}\mathbb{E}\left[\|\Sigma - \hat{\Sigma}\| \cdot \|\hat{w}_{MN}\|^2\right] \geq \lim_{n \to \infty}\left(\sigma^2\frac{n - d_S}{\lambda_n}\right)C\sqrt{\frac{d_S + \lambda_n}{n}} = \infty.$$

It is easy to see that the remaining terms in the lower bound of Proposition B.2 are negligible. $\qquad\square$

### B.3 Uniform convergence on tighter sets (Section 3.2)

**Theorem 3.3.** *In Setting B, let $\mathcal{A}$ be an algorithm outputting interpolators, $X\mathcal{A}(X, Y) = Y$, with*

$$\mathcal{A}\left((X_S, X_J), y\right)_S = \mathcal{A}\left((X_S, -X_J), y\right)_S \quad \text{and} \quad \lim_{n \to \infty}\lim_{d_J \to \infty}L_\mathcal{D}(\mathcal{A}(X, y)) \overset{a.s.}{=} \sigma^2. \quad (6)$$

*For any $\delta \in (0, \frac{1}{2})$ and set of typical training examples $\mathcal{S}_{n,\delta}$ satisfying $\Pr(\mathbf{S} \in \mathcal{S}_{n,\delta}) \geq 1 - \delta$, let $\mathcal{W}_{n,\delta}^\mathcal{A} = \{\mathcal{A}(X, Y) : (X, Y) \in \mathcal{S}_{n,\delta}\}$ denote the set of typical outputs. Then*

$$\lim_{n \to \infty}\lim_{d_J \to \infty}\sup_{\mathbf{S} \in \mathcal{S}_{n,\delta}}\sup_{w \in \mathcal{W}_{n,\delta}}|L_\mathcal{D}(w) - L_\mathbf{S}(w)| \overset{a.s.}{\geq} 3\sigma^2. \quad (7)$$

*Proof.* Fix any $\mathcal{S}_{n,\delta}$ satisfying $\Pr(\mathbf{S} \in \mathcal{S}_{n,\delta}) \geq 1 - \delta$. For each $\mathbf{S} = ((X_S, X_J), Y)$, we define $\tilde{\mathbf{S}} = ((X_S, -X_J), Y)$. Note that the marginal distribution of $\tilde{\mathbf{S}}$ is the same as $\mathbf{S}$ because of the isotropic Gaussian distribution. Thus we also have $\Pr(\tilde{\mathbf{S}} \in \mathcal{S}_{n,\delta}) \geq 1 - \delta$. By a simple union bound

$$1 - \Pr(\mathbf{S} \in \mathcal{S}_{n,\delta} \cap \tilde{\mathbf{S}} \in \mathcal{S}_{n,\delta}) = \Pr(\mathbf{S} \notin \mathcal{S}_{n,\delta} \cup \tilde{\mathbf{S}} \notin \mathcal{S}_{n,\delta})$$
$$\leq \Pr(\mathbf{S} \notin \mathcal{S}_{n,\delta}) + \Pr(\tilde{\mathbf{S}} \notin \mathcal{S}_{n,\delta}) \leq 2\delta.$$

As $\delta < \frac{1}{2}$, we have $\Pr(\mathbf{S} \in \mathcal{S}_{n,\delta} \cap \tilde{\mathbf{S}} \in \mathcal{S}_{n,\delta}) > 0$, so the set $\{\mathbf{S} \in \mathcal{S}_{n,\delta} : \tilde{\mathbf{S}} \in \mathcal{S}_{n,\delta}\}$ must be nonempty. Pick any $\mathbf{S} = ((X_S, X_J), Y)$ in this set; thus $\hat{w} = \mathcal{A}\left((X_S, X_J), Y\right) \in \mathcal{W}_{n,\delta}$ and $\tilde{w} = \mathcal{A}\left((X_S, -X_J), Y\right) \in \mathcal{W}_{n,\delta}$. As $\mathcal{A}$ outputs interpolators, we have that

$$X_S\hat{w}_S + X_J\hat{w}_J = Y = X_S\tilde{w}_S - X_J\tilde{w}_J,$$

and (6) implies that $\hat{w}_S = \tilde{w}_S$, so then $X_J\hat{w}_J = -X_J\tilde{w}_J$. Thus

$$L_\mathbf{S}(\tilde{w}) = \frac{1}{n}\|X\tilde{w} - Y\|^2 = \frac{1}{n}\|X_S\hat{w}_S - X_J\hat{w}_J - (X_S\hat{w}_S + X_J\hat{w}_J)\|^2 = \frac{1}{n}\|-2X_J\hat{w}_J\|^2$$
$$\geq \frac{4}{n}\|(I_n - \Pi)X_J\hat{w}_J\|^2,$$

where $\Pi \in \mathbb{R}^{n \times n}$ is the orthogonal projection onto the range of $X_S$. Now,

$$
\begin{aligned}
(I_n - \Pi) X_J \hat{w}_J &= (I_n - \Pi)(X_S \hat{w}_S + X_J \hat{w}_J) \\
&= (I_n - \Pi) Y \\
&= (I_n - \Pi)(X_S w_S^* + E) \\
&= (I_n - \Pi) E \\
&\sim \mathcal{N}(0, \sigma^2 (I_n - \Pi))
\end{aligned}
$$

using $E \sim \mathcal{N}(0, \sigma^2 I_n)$. As $n \to \infty$, because $X_S$ is almost surely rank $d_S$, $\mathrm{Tr}(I_n - \Pi)$ is almost surely $n - d_S$. Thus we have

$$
\frac{1}{n - d_S} \| (I_n - \Pi) X_J \hat{w}_J \|^2 \overset{a.s.}{\to} \sigma^2,
$$

and so

$$
L_{\mathbf{S}}(\tilde{w}) \overset{a.s.}{\geq} 4\sigma^2 \frac{n - d_S}{n} \to 4\sigma^2.
$$

The conclusion follows by the observation that

$$
\sup_{\mathbf{S} \in \mathcal{S}_{n,\delta}} \sup_{w \in \mathcal{W}_\delta} |L_{\mathcal{D}}(w) - L_{\mathbf{S}}(w)| \geq L_{\mathbf{S}}(\tilde{w}) - L_{\mathcal{D}}(\tilde{w}). \qquad \square
$$

**Proposition B.3.** *Let $f_S : \mathbb{R}^{d_S} \to \mathbb{R}$ and $f_J : \mathbb{R}^{d_J} \to \mathbb{R}$ be convex functions, with $f_J$ symmetric, $f_J(-w) = f_J(w)$. Let $\mathcal{A}$ be an interpolation algorithm satisfying*

$$
\mathcal{A}(X, y) = \underset{w \,\mathrm{s.t.}\, Xw = y}{\arg\min} \ f_S(w_S) + f_J(w_J).
$$

*Then negating junk dimensions simply negates the corresponding dimensions of the predictor:*

$$
\mathcal{A}\left((X_S, -X_J), Y\right) = \begin{bmatrix} I_{d_S} & 0_{d_S \times d_J} \\ 0_{d_J \times d_S} & -I_{d_J} \end{bmatrix} \mathcal{A}\left((X_S, X_J), Y\right).
$$

*(If the minimizer is not unique, the equation holds as an operation on sets.)*

*Proof.* The KKT conditions for $\mathcal{A}(X, y)$, which are both necessary and sufficient in this case, are

$$
Xw = X_S w_S + X_J w_J = Y, \qquad 0 \in \partial f_S(w_S) + \nu_S^\mathsf{T} X_S, \qquad 0 \in \partial f_J(w_J) + \nu_J^\mathsf{T} X_J, \qquad (14)
$$

where $\delta$ denotes the subdifferential, and the dual variables $\nu_S \in \mathbb{R}^{d_S}$ and $\nu_J \in \mathbb{R}^{d_J}$ are otherwise unconstrained. Also note that because $f_J$ is symmetric, if $g \in \partial f_J$ then for any $t$, there is some $g' \in \partial f_J$ such that $g'(-t) = -g(t)$.

Let $(\hat{w}, \nu_S, \nu_J)$ be some solution to (14), and define $\tilde{w} = (\hat{w}_S, -\hat{w}_J)$, $\tilde{X} = (X_S, -X_J)$. Then

$$
(X_S, -X_J)\, \tilde{w} = X_S \tilde{w}_S - X_J \tilde{w}_J = X_S \hat{w}_S + X_J \hat{w}_J = Y,
$$

$$
\partial f_S(\tilde{w}_S) + \nu_S^\mathsf{T} \tilde{X}_S = \partial f_S(\hat{w}_S) + \nu_S^\mathsf{T} X_S \ni 0,
$$

and $\quad \partial f_J(\tilde{w}_J) + \nu_J^\mathsf{T} \tilde{X}_J = \partial f_J(-\hat{w}_J) + \nu_J^\mathsf{T}(-X_J) \ni 0 \quad$ because $\quad 0 \in \partial f_J(\hat{w}_J) + \nu_J^\mathsf{T} X_J.$

Thus $(\tilde{w}, \nu_S, \nu_J)$ satisfies the KKT conditions for $\mathcal{A}(\tilde{X}, Y)$. When the minimizer is not unique, the same argument works in reverse, showing that solution sets are related in the same way. $\qquad \square$

# C  Proofs for Section 4

## C.1  Consistency of the minimal risk interpolator (Proposition 4.3)

**Proposition 4.3.** *In Setting A, the expected risk of the minimal-risk interpolator is*

$$
\mathbb{E}\, L_{\mathcal{D}}(\hat{w}_{MR}) = \frac{p-1}{p-1-n} L_{\mathcal{D}}(w^*).
$$

*Proof.* Recall that

$$\hat{w}_{MR} = w^* + \Sigma^{-1}X^{\mathsf{T}}(X\Sigma^{-1}X^{\mathsf{T}})^{-1}E.$$

From this, we can compute

$$\begin{aligned}
L_{\mathcal{D}}(\hat{w}_{MR}) - L_{\mathcal{D}}(w^*) &= (\hat{w}_{MR} - w^*)^{\mathsf{T}}\Sigma(\hat{w}_{MR} - w^*) \\
&= (\hat{w}_{MR} - w^*)^{\mathsf{T}}X^{\mathsf{T}}(X\Sigma^{-1}X^{\mathsf{T}})^{-1}E \\
&= (X\hat{w}_{MR} - Xw^*)^{\mathsf{T}}(X\Sigma^{-1}X^{\mathsf{T}})^{-1}E \\
&= (Y - Xw^*)^{\mathsf{T}}(X\Sigma^{-1}X^{\mathsf{T}})^{-1}E \\
&= E^{\mathsf{T}}(ZZ^{\mathsf{T}})^{-1}E \\
&= \langle (ZZ^{\mathsf{T}})^{-1}, EE^{\mathsf{T}}\rangle.
\end{aligned}$$

By independence of $Z$ and $E$, we get

$$\mathbb{E}[L_{\mathcal{D}}(\hat{w}_{MR}) - L_{\mathcal{D}}(w^*)] = \sigma^2 \, \mathbb{E}\,\mathrm{Tr}\left[\left(ZZ^{\mathsf{T}}\right)^{-1}\right].$$

Note that $\left(ZZ^{\mathsf{T}}\right)^{-1}$ follows an inverse-Wishart distribution whose expectation is $\frac{I_n}{p-n-1}$. Therefore, we obtain

$$\begin{aligned}
\mathbb{E}[L_{\mathcal{D}}(\hat{w}_{MR})] &= \sigma^2 + \sigma^2 \, \mathrm{Tr}\left(\frac{I_n}{p-n-1}\right) \\
&= \sigma^2\left(1 + \frac{n}{p-n-1}\right) = \left(\frac{p-1}{p-n-1}\right)\cdot L_{\mathcal{D}}(w^*). \qquad \square
\end{aligned}$$

## C.2 Uniform consistency of low norm interpolators (Section 4.1)

### C.2.1 General results

Our key lemma is as follows:

**Lemma C.1.** *Let $\hat{w}$ be any predictor that interpolates the data, with $\|\hat{w}\| \leq B$, and $F \in \mathbb{R}^{p\times(p-n)}$ be the matrix whose columns form an orthonormal basis of the kernel of $X$. In other words, if $X\hat{w} = Y$, $XF = 0_{n\times(p-n)}$ and $F^{\mathsf{T}}F = I_{p-n}$, then (8), the worst-case generalization gap for interpolators up to norm $B$, is equal to*

$$L_{\mathcal{D}}(\hat{w}) + \inf_{\lambda > \|F^{\mathsf{T}}\Sigma F\|}\|F^{\mathsf{T}}[\lambda\hat{w} - \Sigma(\hat{w} - w^*)]\|_{(\lambda I_{p-n} - F^{\mathsf{T}}\Sigma F)^{-1}} + \lambda(B^2 - \|\hat{w}\|^2).$$

*Proof.* Observe that $\{w \in \mathbb{R}^p : L_S(w) = 0\} = \{\hat{w} + Fu : u \in \mathbb{R}^{p-n}\}$. Then

$$\begin{aligned}
&\sup_{\substack{\|w\|\leq B \\ L_S(w)=0}} L_{\mathcal{D}}(w) - L_S(w) \\
&= L_{\mathcal{D}}(w^*) + \sup_{\substack{\|w\|\leq B \\ L_S(w)=0}} L_{\mathcal{D}}(w) - L_{\mathcal{D}}(w^*) \\
&= L_{\mathcal{D}}(w^*) + \sup_{\|\hat{w}+Fu\|^2\leq B^2} (\hat{w} + Fu - w^*)^{\mathsf{T}}\Sigma(\hat{w} + Fu - w^*) \\
&= L_{\mathcal{D}}(w^*) + \sup_{\|u\|^2+2\langle u, F^{\mathsf{T}}\hat{w}\rangle+\|\hat{w}\|^2\leq B^2} u^{\mathsf{T}}(F^{\mathsf{T}}\Sigma F)u + 2\langle u, F^{\mathsf{T}}\Sigma(\hat{w} - w^*)\rangle + (\hat{w} - w^*)^{\mathsf{T}}\Sigma(\hat{w} - w^*) \\
&= L_{\mathcal{D}}(\hat{w}) + \sup_{\|u\|^2+2\langle u, F^{\mathsf{T}}\hat{w}\rangle+\|\hat{w}\|^2\leq B^2} u^{\mathsf{T}}(F^{\mathsf{T}}\Sigma F)u + 2\langle u, F^{\mathsf{T}}\Sigma(\hat{w} - w^*)\rangle \\
&= L_{\mathcal{D}}(\hat{w}) - \inf_{\|u\|^2+2\langle u, F^{\mathsf{T}}\hat{w}\rangle+\|\hat{w}\|^2\leq B^2} u^{\mathsf{T}}(-F^{\mathsf{T}}\Sigma F)u - 2\langle u, F^{\mathsf{T}}\Sigma(\hat{w} - w^*)\rangle.
\end{aligned}$$

Although the second term involves a concave minimization problem, it is a quadratic optimization problem with a single quadratic inequality constraint. This is a classical example where strong duality

holds even though the objective is not convex [9, Appendix B]. In order to derive the dual, we write down the Lagrangian:

$$L(u, \lambda) = u^{\mathsf{T}}(-F^{\mathsf{T}}\Sigma F)u - 2\langle u, F^{\mathsf{T}}\Sigma(\hat{w} - w^*)\rangle + \lambda(\|u\|^2 + 2\langle u, F^{\mathsf{T}}\hat{w}\rangle + \|\hat{w}\|^2 - B^2)$$
$$= u^{\mathsf{T}}(\lambda I_{p-n} - F^{\mathsf{T}}\Sigma F)u + 2\langle u, F^{\mathsf{T}}(\lambda\hat{w} - \Sigma(\hat{w} - w^*))\rangle - \lambda(B^2 - \|\hat{w}\|^2);$$

strong duality tells us that the infimum is equal to $\sup_{\lambda \geq 0} \inf_u L(u, \lambda)$. For $\lambda < \|F^{\mathsf{T}}\Sigma F\|$, $\lambda I_{p-n} - F^{\mathsf{T}}\Sigma F$ has strictly negative eigenvalues, and so then $\inf_u L(u, \lambda) = -\infty$. If instead $\lambda > \|F^{\mathsf{T}}\Sigma F\|$, $\lambda I_{p-n} - F^{\mathsf{T}}\Sigma F$ is strictly positive definite, and setting the $u$ derivative to zero yields that $\inf_u L(\lambda, u)$ is

$$-\left[F^{\mathsf{T}}(\lambda\hat{w} - \Sigma(\hat{w} - w^*))\right]^{\mathsf{T}}(\lambda I_{p-n} - F^{\mathsf{T}}\Sigma F)^{-1}\left[F^{\mathsf{T}}(\lambda\hat{w} - \Sigma(\hat{w} - w^*))\right] - \lambda(B^2 - \|\hat{w}\|^2). \quad (15)$$

If instead $\lambda = \|F^{\mathsf{T}}\Sigma F\|$, we again have $\inf_u L(u, \lambda) = -\infty$ unless $F^{\mathsf{T}}(\lambda\hat{w} - F^{\mathsf{T}}\Sigma(\hat{w} - w^*)) = 0$ so that the linear term is identically zero; in this case, the quadratic term is minimized by $u = 0$, and $\inf_u L(u, \lambda) = \lambda(B^2 - \|\hat{w}\|^2)$ agrees with (15), so this case is covered by the strict case as well. Thus the dual problem is to maximize (15) over $\lambda > \|F^{\mathsf{T}}\Sigma F\|$. The desired result follows by passing the minus sign into the sup of the dual problem. □

We will now prove Theorem 4.5.

**Theorem 4.5.** *The following results hold deterministically, viewing $L_{\mathcal{D}}(w)$ simply as a quadratic function $L_{\mathcal{D}}(w^*) + \|w - w^*\|_{\Sigma}$, with no distributional assumptions on $\mathbf{S}$.*

*(i) It holds that*

$$\sup_{\substack{\|w\| \leq \|\hat{w}_{MR}\| \\ L_{\mathbf{S}}(w)=0}} L_{\mathcal{D}}(w) - L_{\mathbf{S}}(w) = L_{\mathcal{D}}(\hat{w}_{MR}) + \gamma_n\, \kappa_X(\Sigma)\left[\|\hat{w}_{MR}\|^2 - \|\hat{w}_{MN}\|^2\right]$$

*where $1 \leq \gamma_n \leq 4$.*

*If the minimal risk interpolator is consistent, $\mathbb{E}\, L_{\mathcal{D}}(\hat{w}_{MR}) - L_{\mathcal{D}}(w^*) \to 0$, then the class of interpolators with norm less than $\|\hat{w}_{MR}\|$ is uniformly consistent if and only if*

$$\mathbb{E}\, \kappa_X(\Sigma) \cdot \left[\|\hat{w}_{MR}\|^2 - \|\hat{w}_{MN}\|^2\right] \to 0.$$

*(ii) Fix a sequence $(B_n)$ such that $B_n \geq \|\hat{w}_{MN}\|$ for all $n$. Then*

$$\sup_{\|w\| \leq B_n,\, L_{\mathbf{S}}(w)=0} L_{\mathcal{D}}(w) - L_{\mathbf{S}}(w) = L_{\mathcal{D}}(\hat{w}_{MN}) + \kappa_X(\Sigma)\left[B_n^2 - \|\hat{w}_{MN}\|^2\right] + R_n$$

*where $0 \leq R_n \leq 2\sqrt{[L_{\mathcal{D}}(\hat{w}_{MN}) - L_{\mathcal{D}}(w^*)]\, \kappa_X(\Sigma)\, [B_n^2 - \|\hat{w}_{MN}\|^2]}$.*

*If $\mathbb{E}\, L_{\mathcal{D}}(\hat{w}_{MN}) - L_{\mathcal{D}}(w^*) \to 0$, the class of interpolators with norm less than $B_n$ is thus uniformly consistent if and only if*

$$\mathbb{E}\, \kappa_X(\Sigma) \cdot \left[B_n^2 - \|\hat{w}_{MN}\|^2\right] \to 0.$$

*Proof.* For case (i), observe that

$$F^{\mathsf{T}}\Sigma(\hat{w}_{MR} - w^*) = F^{\mathsf{T}}X^{\mathsf{T}}(X\Sigma^{-1}X^{\mathsf{T}})^{-1}E = (XF)^{\mathsf{T}}(X\Sigma^{-1}X^{\mathsf{T}})^{-1}E = 0.$$

Thus picking $\hat{w} = \hat{w}_{MR}$ and $B = \|\hat{w}_{MR}\|$ in Lemma C.1 gives that

$$\sup_{\|w\| \leq \|\hat{w}_{MR}\|,\, L_{\mathbf{S}}(w)=0} L_{\mathcal{D}}(w) = L_{\mathcal{D}}(\hat{w}_{MR}) + \inf_{\lambda > \|F^{\mathsf{T}}\Sigma F\|}\|\lambda F^{\mathsf{T}}\hat{w}_{MR}\|_{(\lambda I_{p-n} - F^{\mathsf{T}}\Sigma F)^{-1}}. \quad (16)$$

Since we have

$$\frac{1}{\lambda}I_{p-n} \preceq (\lambda I_{p-n} - F^{\mathsf{T}}\Sigma F)^{-1},$$

we know that $\sup_{\|w\| \leq \|\hat{w}_{MR}\|,\, L_{\mathbf{S}}(w)=0} L_{\mathcal{D}}(w)$ is lower bounded by

$$L_{\mathcal{D}}(\hat{w}_{MR}) + \inf_{\lambda > \|F^{\mathsf{T}}\Sigma F\|}\frac{1}{\lambda}\|\lambda F^{\mathsf{T}}\hat{w}_{MR}\|^2 = L_{\mathcal{D}}(\hat{w}_{MR}) + \|F^{\mathsf{T}}\Sigma F\| \cdot \|F^{\mathsf{T}}\hat{w}_{MR}\|^2.$$

In order to compute $\|F^\mathsf{T}\hat{w}_{MR}\|^2$, we notice that $FF^\mathsf{T}$ is the orthogonal projection onto the kernel of $X$. Using the fact that $\mathrm{im}(X^\mathsf{T}) = \ker(X)^\perp$, we get $I - FF^\mathsf{T}$ is the orthogonal projection onto the image of $X^\mathsf{T}$. Thus,

$$X(I - FF^\mathsf{T})\hat{w}_{MR} = X\hat{w}_{MR} = Y,$$

and left-multiplying both sides by $X^\mathsf{T}(XX^\mathsf{T})^{-1}$ gives that

$$\hat{w}_{MN} = X^\mathsf{T}(XX^\mathsf{T})^{-1}X(I - FF^\mathsf{T})\hat{w}_{MR} = (I - FF^\mathsf{T})\hat{w}_{MR},$$

and so

$$
\begin{aligned}
\|F^\mathsf{T}\hat{w}_{MR}\|^2 &= \hat{w}_{MR}^\mathsf{T}FF^\mathsf{T}\hat{w}_{MR} \\
&= \hat{w}_{MR}^\mathsf{T}F(F^\mathsf{T}F)F^\mathsf{T}\hat{w}_{MR} \\
&= \|FF^\mathsf{T}\hat{w}_{MR}\|^2 \\
&= \|\hat{w}_{MR}\|^2 - \|(I - FF^\mathsf{T})\hat{w}_{MR}\|^2 \\
&= \|\hat{w}_{MR}\|^2 - \|\hat{w}_{MN}\|^2
\end{aligned}
$$

which establishes the lower bound with a constant of 1.

Similarly, we can use $(\lambda I_{p-n} - F^\mathsf{T}\Sigma F)^{-1} \preceq \frac{1}{\lambda - \|F^\mathsf{T}\Sigma F\|}I_{p-n}$ to upper bound (16) as

$$
\begin{aligned}
& L_\mathcal{D}(\hat{w}_{MR}) + \inf_{\lambda > \|F^\mathsf{T}\Sigma F\|} \frac{1}{\lambda - \|F^\mathsf{T}\Sigma F\|}\|\lambda F^\mathsf{T}\hat{w}_{MR}\|^2 \\
&= L_\mathcal{D}(\hat{w}_{MR}) + \inf_{\lambda > 0} \frac{(\lambda + \|F^\mathsf{T}\Sigma F\|)^2}{\lambda}(\|\hat{w}_{MR}\|^2 - \|\hat{w}_{MN}\|^2) \\
&= L_\mathcal{D}(\hat{w}_{MR}) + \inf_{\lambda > 0} \left(\lambda + 2\|F^\mathsf{T}\Sigma F\| + \frac{\|F^\mathsf{T}\Sigma F\|^2}{\lambda}\right)(\|\hat{w}_{MR}\|^2 - \|\hat{w}_{MN}\|^2) \\
&= L_\mathcal{D}(\hat{w}_{MR}) + 4\|F^\mathsf{T}\Sigma F\| \cdot (\|\hat{w}_{MR}\|^2 - \|\hat{w}_{MN}\|^2).
\end{aligned}
$$

This gives the desired upper bound with a constant of 4. It follows immediately that (16) converges to $L_\mathcal{D}(w^*)$ if and only if

$$\mathbb{E}\|F^\mathsf{T}\Sigma F\| \cdot (\|\hat{w}_{MR}\|^2 - \|\hat{w}_{MN}\|^2) \to 0.$$

Turning to part (ii), observe that

$$F^\mathsf{T}\hat{w}_{MN} = F^\mathsf{T}X^\mathsf{T}(XX^\mathsf{T})^{-1}Y = (XF)^\mathsf{T}(XX^\mathsf{T})^{-1}Y = 0,$$

so that Lemma C.1 with $\hat{w} = \hat{w}_{MN}$ gives

$$\sup_{\|w\| \le B_n, L_\mathbf{S}(w)=0} L_\mathcal{D}(w) = L_\mathcal{D}(\hat{w}_{MN}) + \inf_{\lambda > \|F^\mathsf{T}\Sigma F\|}\|F^\mathsf{T}\Sigma(\hat{w}-w^*)\|_{(\lambda I_{p-n} - F^\mathsf{T}\Sigma F)^{-1}} + \lambda(B_n^2 - \|\hat{w}_{MN}\|^2).$$

Moreover, it is clear that

$$0_{p-n} \prec (\lambda I_{p-n} - F^\mathsf{T}\Sigma F)^{-1} \prec \frac{1}{\lambda - \|F^\mathsf{T}\Sigma F\|}I_{p-n}.$$

Therefore, $\sup_{\|w\| \le B_n, L_\mathbf{S}(w)=0} L_\mathcal{D}(w)$ is lower bounded by, recalling that $\|F^\mathsf{T}\Sigma F\| = \kappa_X(\Sigma)$,

$$L_\mathcal{D}(\hat{w}_{MN}) + \inf_{\lambda > \|F^\mathsf{T}\Sigma F\|} \lambda(B_n^2 - \|\hat{w}_{MN}\|) = L_\mathcal{D}(\hat{w}_{MN}) + \kappa_X(\Sigma) \cdot \left[B_n^2 - \|\hat{w}_{MN}\|^2\right], \quad (17)$$

and we have shown that $R_n \ge 0$ in the result. On the other hand, $\sup_{\|w\| \le B_n, L_\mathbf{S}(w)=0} L_\mathcal{D}(w)$ is upper bounded by

$$
\begin{aligned}
& L_\mathcal{D}(\hat{w}_{MN}) + \inf_{\lambda > \|F^\mathsf{T}\Sigma F\|} \frac{1}{\lambda - \|F^\mathsf{T}\Sigma F\|}\|F^\mathsf{T}\Sigma(\hat{w}_{MN} - w^*)\|^2 + \lambda\left[B_n^2 - \|\hat{w}_{MN}\|^2\right] \\
&= L_\mathcal{D}(\hat{w}_{MN}) + \inf_{\lambda > 0} \frac{1}{\lambda}\|F^\mathsf{T}\Sigma(\hat{w}_{MN} - w^*)\|^2 + (\lambda + \kappa_X(\Sigma))\left[B_n^2 - \|\hat{w}_{MN}\|^2\right] \\
&= L_\mathcal{D}(\hat{w}_{MN}) + \kappa_X(\Sigma) \cdot \left[B_n^2 - \|\hat{w}_{MN}\|^2\right] + \inf_{\lambda > 0} \frac{1}{\lambda}\|F^\mathsf{T}\Sigma(\hat{w}_{MN} - w^*)\|^2 + \lambda\left[B_n^2 - \|\hat{w}_{MN}\|^2\right] \\
&= L_\mathcal{D}(\hat{w}_{MN}) + \kappa_X(\Sigma) \cdot \left[B_n^2 - \|\hat{w}_{MN}\|^2\right] + 2\sqrt{\|F^\mathsf{T}\Sigma(\hat{w}_{MN} - w^*)\|^2 \cdot \left[B_n^2 - \|\hat{w}_{MN}\|^2\right]}. \quad (18)
\end{aligned}
$$

We can upper bound

$$\|F^{\mathsf{T}}\Sigma(\hat{w}_{MN} - w^*)\|^2 = (\hat{w}_{MN} - w^*)^{\mathsf{T}}\Sigma F F^{\mathsf{T}}\Sigma(\hat{w}_{MN} - w^*)$$
$$= [\Sigma^{1/2}(\hat{w}_{MN} - w^*)]^{\mathsf{T}}(\Sigma^{1/2}FF^{\mathsf{T}}\Sigma^{1/2})[\Sigma^{1/2}(\hat{w}_{MN} - w^*)]$$
$$\leq \|\Sigma^{1/2}FF^{\mathsf{T}}\Sigma^{1/2}\| \cdot \|\Sigma^{1/2}(\hat{w}_{MN} - w^*)\|^2$$
$$= \|F^{\mathsf{T}}\Sigma F\| \cdot [L_{\mathcal{D}}(\hat{w}_{MN}) - L_{\mathcal{D}}(w^*)]$$

using the fact that $\|AA^T\| = \|A^T A\|$ with $A = F^T\Sigma^{1/2}$. Plugging into the third term of (18) yields our desired upper bound on $R_n$,

To show the statement about expectations when $\mathbb{E}\,L_{\mathcal{D}}(\hat{w}_{MN}) - L_{\mathcal{D}}(w^*) \to 0$, note for one direction that (17) gives

$$\liminf_{n\to\infty}\mathbb{E}\left[\sup_{\substack{\|w\|\leq B_n \\ L_{\mathbf{S}}(w)=0}} L_{\mathcal{D}}(w) - L_{\mathbf{S}}(w)\right] \geq L_{\mathcal{D}}(w^*) + \lim_{n\to\infty}\mathbb{E}\,\kappa_X(\Sigma) \cdot \left[B_n^2 - \|\hat{w}_{MN}\|^2\right].$$

For the other direction, we have

$$R_n \leq 2\sqrt{\|F^{\mathsf{T}}\Sigma F\| \cdot [L_{\mathcal{D}}(\hat{w}_{MN}) - L_{\mathcal{D}}(w^*)]\left[B_n^2 - \|\hat{w}_{MN}\|^2\right]}$$
$$\leq \epsilon\|F^{\mathsf{T}}\Sigma F\| \cdot \left[B_n^2 - \|\hat{w}_{MN}\|^2\right] + \frac{1}{\epsilon}[L_{\mathcal{D}}(\hat{w}_{MN}) - L_{\mathcal{D}}(w^*)]$$

for any $\epsilon > 0$. This implies

$$\limsup_{n\to\infty}\mathbb{E}\left[\sup_{\substack{\|w\|\leq B_n \\ L_{\mathbf{S}}(w)=0}} L_{\mathcal{D}}(w) - L_{\mathbf{S}}(w)\right] \leq L_{\mathcal{D}}(w^*) + (1+\epsilon)\,\mathbb{E}\left(\lim_{n\to\infty}\kappa_X(\Sigma) \cdot \left[B_n^2 - \|\hat{w}_{MN}\|^2\right]\right),$$

showing the desired result. □

### C.2.2 Special case of Setting B

In Setting **B**, we are able to compute $\kappa_X(\Sigma)$.

**Proposition C.2.** *With probability 1, it holds in Setting **B** that*

$$\lim_{d_J\to\infty}\kappa_X(\Sigma) = \frac{\lambda_n}{n}\left\|\left[\frac{X_S^{\mathsf{T}}X_S}{n} + \frac{\lambda_n}{n}I_{d_S}\right]^{-1}\right\|.$$

*Proof.* Recall that

$$\kappa_X(\Sigma) = \|F^{\mathsf{T}}\Sigma F\| = \|\Sigma^{1/2}FF^{\mathsf{T}}\Sigma^{1/2}\| = \|\Sigma^{1/2}(I - X^{\mathsf{T}}(XX^{\mathsf{T}})^{-1}X)\Sigma^{1/2}\|.$$

It is a routine calculation to show that

$$\Sigma^{1/2}FF^{\mathsf{T}}\Sigma^{1/2} = \begin{bmatrix} I_{d_S} - X_S^{\mathsf{T}}(X_SX_S^{\mathsf{T}} + X_JX_J^{\mathsf{T}})^{-1}X_S & -\sqrt{\frac{\lambda_n}{d_J}}X_S^{\mathsf{T}}(X_SX_S^{\mathsf{T}} + X_JX_J^{\mathsf{T}})^{-1}X_J \\ -\sqrt{\frac{\lambda_n}{d_J}}X_J^{\mathsf{T}}(X_SX_S^{\mathsf{T}} + X_JX_J^{\mathsf{T}})^{-1}X_S & \frac{\lambda_n}{d_J}\left[I_{d_J} - X_J^{\mathsf{T}}(X_SX_S^{\mathsf{T}} + X_JX_J^{\mathsf{T}})^{-1}X_J\right] \end{bmatrix}.$$

Intuitively, since only the upper-left block does not vanish as $d_J \to \infty$, we should expect

$$\lim_{d_J\to\infty}\kappa_X(\Sigma) = \|I_{d_S} - X_S^{\mathsf{T}}(X_SX_S^{\mathsf{T}} + \lambda_n I_n)^{-1}X_S\|.$$

However, as the dimensions of $\Sigma^{1/2}FF^{\mathsf{T}}\Sigma^{1/2}$ also increase with $d_J$, the analysis of $\kappa_X(\Sigma)$ requires more care.

It is clear that $\kappa_X(\Sigma) \geq \|I_{d_S} - X_S^{\mathsf{T}}(X_SX_S^{\mathsf{T}} + X_JX_J^{\mathsf{T}})^{-1}X_S\|$, and so

$$\liminf_{d_J\to\infty}\kappa_X(\Sigma) \geq \|I_{d_S} - X_S^{\mathsf{T}}(X_SX_S^{\mathsf{T}} + \lambda_n I_n)^{-1}X_S\|.$$

To upper bound the limit, fix any $v = (v_1, v_2)$ such that $v_1 \in \mathbb{R}^{d_S}$, $v_2 \in \mathbb{R}^{d_J}$ and $\|v\| = 1$. We can write

$$
\begin{aligned}
v^\mathsf{T}\Sigma^{1/2}FF^\mathsf{T}\Sigma^{1/2}v &= v_1^\mathsf{T}(I_{d_S} - X_S^\mathsf{T}(X_S X_S^\mathsf{T} + X_J X_J^\mathsf{T})^{-1}X_S)v_1 \\
&\quad + \frac{\lambda_n}{d_J}v_2^\mathsf{T}\left[I_{d_J} - X_J^\mathsf{T}(X_S X_S^\mathsf{T} + X_J X_J^\mathsf{T})^{-1}X_J\right]v_2 \\
&\quad - 2\sqrt{\frac{\lambda_n}{d_J}}v_1^\mathsf{T}X_S^\mathsf{T}(X_S X_S^\mathsf{T} + X_J X_J^\mathsf{T})^{-1}X_J v_2.
\end{aligned}
\tag{19}
$$

The first term is upper bounded by

$$
\|I_{d_S} - X_S^\mathsf{T}(X_S X_S^\mathsf{T} + X_J X_J^\mathsf{T})^{-1}X_S\| \cdot \|v_1\| \le \|I_{d_S} - X_S^\mathsf{T}(X_S X_S^\mathsf{T} + X_J X_J^\mathsf{T})^{-1}X_S\|,
$$

and the second term is upper bounded by $\lambda_n/d_J$, because

$$
v_2^\mathsf{T}v_2 \le 1 \qquad \text{and} \qquad v_2^\mathsf{T}X_J^\mathsf{T}(X_S X_S^\mathsf{T} + X_J X_J^\mathsf{T})^{-1}X_J v_2 \ge 0.
$$

For any $\epsilon > 0$, we have

$$
\begin{aligned}
&-2\sqrt{\frac{\lambda_n}{d_J}}v_1^\mathsf{T}X_S^\mathsf{T}(X_S X_S^\mathsf{T} + X_J X_J^\mathsf{T})^{-1}X_J v_2 \\
&\le 2\|v_1^\mathsf{T}X_S^\mathsf{T}(X_S X_S^\mathsf{T} + X_J X_J^\mathsf{T})^{-1/2}\| \cdot \left\|\sqrt{\frac{\lambda_n}{d_J}}(X_S X_S^\mathsf{T} + X_J X_J^\mathsf{T})^{-1/2}X_J v_2\right\| \\
&\le \epsilon\|v_1^\mathsf{T}X_S^\mathsf{T}(X_S X_S^\mathsf{T} + X_J X_J^\mathsf{T})^{-1/2}\|^2 + \frac{1}{\epsilon}\left\|\sqrt{\frac{\lambda_n}{d_J}}(X_S X_S^\mathsf{T} + X_J X_J^\mathsf{T})^{-1/2}X_J v_2\right\|^2 \\
&\le \epsilon\|X_S^\mathsf{T}(X_S X_S^\mathsf{T} + X_J X_J^\mathsf{T})^{-1}X_S\| + \frac{\lambda_n}{\epsilon\, d_J}\|X_J^\mathsf{T}(X_S X_S^\mathsf{T} + X_J X_J^\mathsf{T})^{-1}X_J\| \\
&= \epsilon\|X_S^\mathsf{T}(X_S X_S^\mathsf{T} + X_J X_J^\mathsf{T})^{-1}X_S\| + \frac{\lambda_n}{\epsilon\, d_J}\|(X_S X_S^\mathsf{T} + X_J X_J^\mathsf{T})^{-1/2}X_J X_J^\mathsf{T}(X_S X_S^\mathsf{T} + X_J X_J^\mathsf{T})^{-1/2}\|.
\end{aligned}
$$

Taking a supremum over $v$ in (19), we get

$$
\begin{aligned}
\kappa_X(\Sigma) &\le \|I_{d_S} - X_S^\mathsf{T}(X_S X_S^\mathsf{T} + X_J X_J^\mathsf{T})^{-1}X_S\| + \epsilon\|X_S^\mathsf{T}(X_S X_S^\mathsf{T} + X_J X_J^\mathsf{T})^{-1}X_S\| \\
&\quad + \frac{\lambda_n}{d_J}\left[1 + \frac{1}{\epsilon}\|(X_S X_S^\mathsf{T} + X_J X_J^\mathsf{T})^{-1/2}X_J X_J^\mathsf{T}(X_S X_S^\mathsf{T} + X_J X_J^\mathsf{T})^{-1/2}\|\right].
\end{aligned}
$$

Note that

$$
\begin{aligned}
\lim_{d_J \to \infty}\|(X_S X_S^\mathsf{T} + X_J X_J^\mathsf{T})^{-1/2}X_J X_J^\mathsf{T}(X_S X_S^\mathsf{T} + X_J X_J^\mathsf{T})^{-1/2}\| \\
= \lambda_n\|(X_S X_S^\mathsf{T} + \lambda_n I_n)^{-1}\| < \infty,
\end{aligned}
$$

so for any $\epsilon > 0$,

$$
\limsup_{d_J \to \infty} \kappa_X(\Sigma) \le \|I_{d_S} - X_S^\mathsf{T}(X_S X_S^\mathsf{T} + \lambda_n I_n)^{-1}X_S\| + \epsilon\|X_S^\mathsf{T}(X_S X_S^\mathsf{T} + \lambda_n I_n)^{-1}X_S\|.
$$

Sending $\epsilon \to 0$ matches the $\liminf$ and $\limsup$. Finally, because

$$
(X_S X_S^\mathsf{T} + \lambda_n I_n)^{-1}X_S = X_S(X_S^\mathsf{T}X_S + \lambda_n I_{d_S})^{-1},
$$

we have

$$
\begin{aligned}
I_{d_S} - X_S^\mathsf{T}(X_S X_S^\mathsf{T} + \lambda_n I_n)^{-1}X_S &= I_{d_S} - X_S^\mathsf{T}X_S(X_S^\mathsf{T}X_S + \lambda_n I_{d_S})^{-1} \\
&= \lambda_n(X_S^\mathsf{T}X_S + \lambda_n I_{d_S})^{-1} \\
&= \frac{\lambda_n}{n}\left[\frac{X_S^\mathsf{T}X_S}{n} + \frac{\lambda_n}{n}I_{d_S}\right]^{-1}
\end{aligned}
$$

and the proof is concluded. $\qquad\square$

**Proposition C.3.** *In Setting **B**, it holds that*

$$\lim_{n\to\infty}\lim_{d_J\to\infty}\mathbb{E}\,\kappa_X(\Sigma)\cdot\|\hat{w}_{MN}\|^2 = L_\mathcal{D}(w^*),$$

$$\lim_{n\to\infty}\lim_{d_J\to\infty}\mathbb{E}\,\kappa_X(\Sigma)\cdot\left[\|\hat{w}_{MR}\|^2-\|\hat{w}_{MN}\|^2\right] = 0.$$

*Proof.* Notice that $\kappa_X(\Sigma)\cdot\|\hat{w}_{MN}\|^2$ can be dominated by $\|\Sigma\|\cdot\|\hat{w}_{MR}\|^2$ and Proposition B.1 showed that $\|\hat{w}_{MR}\|^2$ is integrable, so by the dominated convergence theorem,

$$\lim_{d_J\to\infty}\mathbb{E}\,\kappa_X(\Sigma)\cdot\|\hat{w}_{MN}\|^2 = \mathbb{E}\lim_{d_J\to\infty}\kappa_X(\Sigma)\cdot\|\hat{w}_{MN}\|^2.$$

Similarly, $\lim_{d_J\to\infty}\kappa_X(\Sigma)\cdot\|\hat{w}_{MN}\|^2$ can be dominated by

$$\lim_{d_J\to\infty}\kappa_X(\Sigma)\cdot\|\hat{w}_{MR}\|^2 \overset{a.s.}{=} \frac{\lambda_n}{n}\left\|\left[\frac{X_S^\mathsf{T}X_S}{n}+\frac{\lambda_n}{n}I_{d_S}\right]^{-1}\right\|\cdot\left(\|w^*\|^2+\frac{\|E\|^2}{\lambda_n}\right)$$

according to Propositions B.1 and C.2.

As $\left\|\left[\frac{X_S^\mathsf{T}X_S}{n}+\frac{\lambda_n}{n}I_{d_S}\right]^{-1}\right\|\overset{a.s.}{\to}1$ and $\frac{\|E\|^2}{n}\overset{a.s.}{\to}\sigma^2$, we have

$$\lim_{n\to\infty}\lim_{d_J\to\infty}\kappa_X(\Sigma)\cdot\|\hat{w}_{MR}\|^2\overset{a.s.}{=}\sigma^2.$$

Moreover, by independence of $X_S$ and $E$

$$\mathbb{E}\lim_{d_J\to\infty}\kappa_X(\Sigma)\cdot\|\hat{w}_{MR}\|^2 = \left(\frac{\lambda_n\|w^*\|^2}{n}+\sigma^2\right)\cdot\mathbb{E}\left\|\left[\frac{X_S^\mathsf{T}X_S}{n}+\frac{\lambda_n}{n}I_{d_S}\right]^{-1}\right\|.$$

Again, $\left\|\left[\frac{X_S^\mathsf{T}X_S}{n}+\frac{\lambda_n}{n}I_{d_S}\right]^{-1}\right\|$ can be dominated by $\mathrm{Tr}\left(\left(\frac{X_S^\mathsf{T}X_S}{n}\right)^{-1}\right)$, so that

$$\lim_{n\to\infty}\mathbb{E}\lim_{d_J\to\infty}\kappa_X(\Sigma)\cdot\|\hat{w}_{MR}\|^2 = \sigma^2 = \mathbb{E}\lim_{n\to\infty}\lim_{d_J\to\infty}\kappa_X(\Sigma)\cdot\|\hat{w}_{MR}\|^2.$$

It is also straightforward to check that

$$\lim_{n\to\infty}\mathbb{E}\frac{\lambda_n}{n}\left(\lim_{d_J\to\infty}\|\hat{w}_{MR}\|^2\right) = \sigma^2 = \mathbb{E}\lim_{n\to\infty}\frac{\lambda_n}{n}\cdot\left(\lim_{d_J\to\infty}\|\hat{w}_{MR}\|^2\right).$$

Another application of DCT shows that

$$\begin{aligned}
\lim_{n\to\infty}\lim_{d_J\to\infty}\mathbb{E}\,\kappa_X(\Sigma)\cdot\|\hat{w}_{MN}\|^2 &= \lim_{n\to\infty}\mathbb{E}\lim_{d_J\to\infty}\kappa_X(\Sigma)\cdot\|\hat{w}_{MN}\|^2\\
&= \mathbb{E}\lim_{n\to\infty}\lim_{d_J\to\infty}\kappa_X(\Sigma)\cdot\|\hat{w}_{MN}\|^2\\
&= \mathbb{E}\lim_{n\to\infty}\frac{\lambda_n}{n}\left\|\left[\frac{X_S^\mathsf{T}X_S}{n}+\frac{\lambda_n}{n}I_{d_S}\right]^{-1}\right\|\cdot\left(\lim_{d_J\to\infty}\|\hat{w}_{MN}\|^2\right)\\
&= \mathbb{E}\lim_{n\to\infty}\frac{\lambda_n}{n}\cdot\left(\lim_{d_J\to\infty}\|\hat{w}_{MN}\|^2\right).
\end{aligned}$$

Using the fact that

$$\frac{\lambda_n}{n}\cdot\left(\lim_{d_J\to\infty}\|\hat{w}_{MN}\|^2\right) \leq \frac{\lambda_n}{n}\cdot\left(\lim_{d_J\to\infty}\|\hat{w}_{MR}\|^2\right)$$

and $\|\hat{w}_{MN}\|^2\leq\|\hat{w}_{MR}\|^2$, two final applications of DCT give

$$\begin{aligned}
\lim_{n\to\infty}\lim_{d_J\to\infty}\mathbb{E}\,\kappa_X(\Sigma)\cdot\|\hat{w}_{MN}\|^2 &= \lim_{n\to\infty}\frac{\lambda_n}{n}\left(\mathbb{E}\lim_{d_J\to\infty}\|\hat{w}_{MN}\|^2\right)\\
&= \lim_{n\to\infty}\frac{\lambda_n}{n}\left(\lim_{d_J\to\infty}\mathbb{E}\|\hat{w}_{MN}\|^2\right)\\
&= \lim_{n\to\infty}\frac{\lambda_n}{n}\left[\|w^*\|^2+\sigma^2\frac{n-d_S}{\lambda_n}+\beta_n\left(\frac{\sigma^2 d_S-\lambda_n\|w_S^*\|^2}{n}\right)\right]\\
&= \sigma^2.
\end{aligned}$$

by Proposition B.1. Consequently, we have established
$$\lim_{n \to \infty} \lim_{d_J \to \infty} \mathbb{E} \left[ \kappa_X(\Sigma) \cdot \left( \|\hat{w}_{MR}\|^2 - \|\hat{w}_{MN}\|^2 \right) \right] = 0. \qquad \square$$

We are finally ready to prove Theorem 4.1 and Proposition 4.6.

**Proposition 4.6.** *In Setting* **B** *with* $\lambda_n = o(n)$,
$$\lim_{n \to \infty} \lim_{d_J \to \infty} \mathbb{E} \left[ \sup_{\|w\| \le \|\hat{w}_{MR}\|, \, L_{\mathbf{S}}(w)=0} L_{\mathcal{D}}(w) - L_{\mathbf{S}}(w) \right] = L_{\mathcal{D}}(w^*).$$

*Proof.* Recall in the proof of Theorem 4.5, it is shown that
$$\sup_{\|w\| \le \|\hat{w}_{MR}\|, \, L_{\mathbf{S}}(w)=0} L_{\mathcal{D}}(w) \le L_{\mathcal{D}}(\hat{w}_{MR}) + 4 \, \kappa_X(\Sigma) \cdot \left[ \|\hat{w}_{MR}\|^2 - \|\hat{w}_{MN}\|^2 \right].$$
Proposition 4.3 implies that
$$\lim_{d_J \to \infty} \mathbb{E} \, L_{\mathcal{D}}(\hat{w}_{MR}) = L_{\mathcal{D}}(w^*).$$
Combined with Proposition C.3, we have shown
$$\lim_{n \to \infty} \lim_{d_J \to \infty} \mathbb{E} \left[ \sup_{\|w\| \le \|\hat{w}_{MR}\|, \, L_{\mathbf{S}}(w)=0} L_{\mathcal{D}}(w) - L_{\mathbf{S}}(w) \right] \le L_{\mathcal{D}}(w^*).$$
On the other hand, we have the trivial lower bound
$$\lim_{n \to \infty} \lim_{d_J \to \infty} \mathbb{E} \left[ \sup_{\substack{\|w\| \le \|\hat{w}_{MR}\| \\ L_{\mathbf{S}}(w)=0}} L_{\mathcal{D}}(w) - L_{\mathbf{S}}(w) \right] \ge \lim_{n \to \infty} \lim_{d_J \to \infty} \mathbb{E} \, L_{\mathcal{D}}(\hat{w}_{MR}) = L_{\mathcal{D}}(w^*). \qquad \square$$

**Theorem 4.1.** *In Setting* **B** *with* $\lambda_n = o(n)$, *fix a sequence* $(\alpha_n) \to \alpha$, *with each* $\alpha_n \ge 1$. *Then*
$$\lim_{n \to \infty} \lim_{d_J \to \infty} \mathbb{E} \left[ \sup_{\|w\| \le \alpha_n \|\hat{w}_{MN}\|, \, L_{\mathbf{S}}(w)=0} L_{\mathcal{D}}(w) - L_{\mathbf{S}}(w) \right] = \alpha^2 L_{\mathcal{D}}(w^*).$$

*Proof.* In the proof of Theorem 4.5, it is shown for every $\epsilon \ge 0$ that
$$\sup_{\substack{\|w\| \le B_n \\ L_{\mathbf{S}}(w)=0}} L_{\mathcal{D}}(w) - L_{\mathbf{S}}(w) \le L_{\mathcal{D}}(\hat{w}_{MN}) + (1+\epsilon) \kappa_X(\Sigma) \cdot \left[ B_n^2 - \|\hat{w}_{MN}\|^2 \right] + \frac{1}{\epsilon} [L_{\mathcal{D}}(\hat{w}_{MN}) - L_{\mathcal{D}}(w^*)].$$
Proposition 4.6 implies that $\lim_{n \to \infty} \lim_{d_J \to \infty} \mathbb{E} \, L_{\mathcal{D}}(\hat{w}_{MN}) = L_{\mathcal{D}}(w^*)$. Thus, plugging in $B_n = \alpha_n \|\hat{w}_{MN}\|$ and taking expectations and limits on both sides gives
$$\lim_{n \to \infty} \lim_{d_J \to \infty} \mathbb{E} \left[ \sup_{\substack{\|w\| \le \alpha_n \|\hat{w}_{MN}\| \\ L_{\mathbf{S}}(w)=0}} L_{\mathcal{D}}(w) \right] \le L_{\mathcal{D}}(w^*) + (1+\epsilon) \lim_{n \to \infty} \lim_{d_J \to \infty} \mathbb{E}(\alpha_n^2 - 1) \kappa_X(\Sigma) \|\hat{w}_{MN}\|^2;$$
further applying Proposition C.3 yields
$$\lim_{n \to \infty} \lim_{d_J \to \infty} \mathbb{E} \left[ \sup_{\substack{\|w\| \le \alpha_n \|\hat{w}_{MN}\| \\ L_{\mathbf{S}}(w)=0}} L_{\mathcal{D}}(w) \right] \le L_{\mathcal{D}}(w^*) + (1+\epsilon)(\alpha^2 - 1) L_{\mathcal{D}}(w^*).$$
Sending $\epsilon \to 0$ yields the upper bound $\alpha^2 L_{\mathcal{D}}(w^*)$.

To get the lower bound, in the proof of Theorem 4.5 it is also shown
$$\sup_{\substack{\|w\| \le B_n \\ L_{\mathbf{S}}(w)=0}} L_{\mathcal{D}}(w) - L_{\mathbf{S}}(w) \ge L_{\mathcal{D}}(\hat{w}_{MN}) + \kappa_X(\Sigma) \cdot \left[ B_n^2 - \|\hat{w}_{MN}\|^2 \right].$$
By Proposition C.3, letting $B_n = \alpha_n \|\hat{w}_{MN}\|$ we obtain
$$\lim_{n \to \infty} \lim_{d_J \to \infty} \mathbb{E} \left[ \sup_{\|w\| \le \alpha_n \|\hat{w}_{MN}\|, \, L_{\mathbf{S}}(w)=0} L_{\mathcal{D}}(w) \right] \ge L_{\mathcal{D}}(w^*) + (\alpha^2 - 1) L_{\mathcal{D}}(w^*) = \alpha^2 L_{\mathcal{D}}(w^*)$$
and the proof is concluded. $\qquad \square$