[Reviews · NeurIPS 2020]

Review 1

Summary and Contributions: The paper shows examples for which the traditional uniform convergence (UC) definition suffers from a large gap compared to other generalization methods. In contrast, the paper shows that an alternative data-based UC definition may tightly control the generalization of those examples. The authors raise questions about the efficiency of data-independent UC-based methods to bound generalization in general undetermined problems.

Strengths: The authors question the efficiency of a method in the heart of the theoretical work in the machine learning field. Focusing on this question might help to a future understanding of generalization in several models. The paper suggests an alternative approach to bound generalization that may produce better bounds in the future. The paper is well written and well explain its results and their contributions.

Weaknesses: The paper's definitions of "consistency" and "interpolation learning" were missing, and explained by the authors during the rebuttal phase. I would suggest to add those definitions should to the paper itself.

Correctness: The proofs/sketches inside the paper seem correct. I did not review the proofs in the appendix.

Clarity: The paper is generally well written and organized.

Relation to Prior Work: The authors clearly explained their contribution over previous works.

Reproducibility: Yes

Additional Feedback:


Review 2

Summary and Contributions: The paper studies a noisy linear regression model to see if a uniform convergence based proof can recover consistency of the minimum norm solution. The authors show that 1. uniform convergence over small norm balls cannot recover the consistency result, nor do any two-sided uniform convergence. 2. however, uniform convergence over small norm balls constrained to the training error being zero, does suffice to show consistency. --------------- Post author feedback comments --------------- I thank the reviewers for the response. After reading the other reviews and discussion the other reviewers, I changed my score to an 8. The primary reason for this is that the authors don't compare their setting with those in the (recent) literature of overparamterized linear regression. I hope that the authors provide a discussion to this end in the revision, if accepted.

Strengths: The paper studies a natural and important problem and makes strong theoretical contributions. All the results are presented and proved very rigorously. Moreover, the authors attempt to guide the reader with the underlying intuitive reasoning. The intermediate discussions the authors have, provide interesting insights as well. For example, after equation 9, the authors say that the result also gives generalization error of interpolators larger than minimal norm, which is an interesting result. The other one is the discussion on (existing) optimistic rates and why they are insufficient. All in all, given the recent interest in interpolation learning and failure(?) of uniform convergence in some settings, the problem that the authors choose to study is timely, and the authors provide strong and insightful results which can influence future investigations.

Weaknesses: 1 (a). Writing: The first complaint is with regard to writing/presentation, which is fixable. Most of the technical meat in the main paper is written like a story, which is helpful to guide the reader, however, the intermediate reasoning is perhaps not as clear (to me) as the authors think it is - it took me multiple passes through the text, as well as look at the proofs, to understand it well. Moreover, sometimes, the authors "hand-wave" the steps: or example, in line 113, they say the "loss is effectively $2(T_n+Q_n)$-Lipschitz" - what is "effectively"? Furthermore, they say that $T_n, Q_n$ goes to $0$ for large $n$- why? I don't doubt the veracity of these claims, but a little more justification would be useful to the reader. 1 (b). Organization of proofs: Although the proofs are written rigorously, in some parts I felt that it makes references to observations which were proved in other propositions (which the reader would not have read upto this point). For example: proof of proposition 3.1 starts with a reference to Proposition C.3, and in some places in Proposition C.3, I felt that it is not justified enough: for example - line 538 where $E{Tr(ZZ^\top)/p} = n/p(p-n-1)$ - this calculation/result appears without any justification, and I later found that it is explained in the proof of Proposition 4.3. Another example is in proof of Theorem 3.2, in the first equation, the term $\hat \Sigma$ appears which is undefined (as of now). Moreover, it is not (immediately) clear why we want to prove the first line. Only at the end of the proof, I see that it refers to Proposition B.1 where $\hat \Sigma$ is defined as well. There is perhaps an order in which the authors wrote these propositions and have in mind for the readers to read it, but it is unspecified. 2. Junk feature setting (not a weakness, but perhaps some more discussion would help): The results are obtained in a setting which the authors call the junk feature setting. Here, instead of the number of parameters $p$ goes $\infty$ (as a function of $n$), a subset of features, called the junk features, go to $\infty$, even for finitely many samples. The interesting part is that in the limit of infinite junk features, the solution behaves like the Ridge regression solution on the finitely many "signal" features. The authors remark that this simple setting is appealing because it captures a lot of interesting properties of interpolaters, especially the double-descent phenomenon. However, in this setting, even though the number of features are increasing, (allowing interpolation), the effective number of parameters, which make the prediction on test data, is fixed (even for infinite $n$). So are these assumptions on the distribution, a "trick" to make the ridge regression solution interpolate on training data, and we can still use nice properties of the ridge regression solution? However, clearly the author do not simply rely on known properties of the ridge regression solution and derive a lot of interesting results. Therefore, more discussion on to this intentional similarity to ridge regression solution would be useful.

Correctness: I skimmed through the proofs and I could not find anything that seemed incorrect. Moreover, the intermediate steps in the proofs are well explained.

Clarity: The motivation is very clear, and the authors contrast the work with related work well. The technical content, at times, felt heavy and rushed to me (more on this in weakness 1) Typos: line 80: in the equation below, is w in $R^p$, or $R^{d_s}$? line 125: $\hat \Sigma$ appears in proof sketch of Thm 3.2, which is undefined yet.

Relation to Prior Work: Related work, although discussed concisely, is clear and the problem studied is put well in context to them.

Reproducibility: Yes

Additional Feedback: Some places where I think more discussion can help: 1. Relation between ridge regression and junk features setting (see weakness 2) 2. There is no discussion on why minimal risk solution is a useful tool for the proofs. I see it arising in Theorem 4.5 and in proof sketches, but is their an intuitive justification for its appearance in the bounds, or is it an artifact of the proof technique? 3. The restricted eigenvalue under Covariance .. $\kappa_X(\Sigma)$ term is hardly discussed - what it means, and why is it important. 4. One parenthetical remark, which I did not understand is the one that authors make in line 165: "Again, since one-sided uniform convergence is always a consequence of consistency, this question is essentially one of viewpoint: do you first show uniform convergence and then bound consistency through uniform convergence, or do you establish uniform convergence as a consequence of consistency?" Why is one-sided uniform convergence "always" a consequence of consistency? - would be helpful if the authors expand upon it.


Review 3

Summary and Contributions: 1. The authors analyze over parameterized linear regression, particularly when it's possible to be risk consistent when dimension goes to infinity for any fixed sample size, and the sample size tends to infinity. 2. The scope of this analysis is broader than previous work in that it does not only consider the min-norm ERM, but also ``small''-norm ERMs 3. (?) The authors introduce ``restricted'' uniform bounds (a la Eq.~(7)) for interpolating predictors, to reduce the complexity component relative to a traditional uniform bound.

Strengths: 1. The claims all seem to be correct 2. The new avenue of ``small'' norm ERMs is quite interesting, and can likely be extended to consideration in other problems. 3. The idea of ``restricted'' uniform bounds for interpolating learners is quite interesting, and a promising avenue of research for more complex settings.

Weaknesses: 1. There is limited discussion of what components of the analysis are bespoke to over parameterized linear regression, and how the present work might inspire similar techniques for other, possibly more complex, problems. I think this may impact the longer term impact of this work. I think the work would benefit from further consideration and discussion of what the novel and ``portable'' insights were. 2. The conditions required in Thm.~4.5 (the main result) for consistency are somewhat obfuscated. The conditions involving $E[\kappa_X(\Sigma) (\norm{\hat w_{MR}} - \norm{\hat w_{MN}}) ]$ should be expressible as conditions on the (sequence of) covariance matrices. In the way they are currently expressed, it is difficult to compare with the ``benign'' condition of BLLT'20 [3] or the ``weakly benign'' condition of NDR'20 [23]. It would be useful if there was a more direct comparison with existing work here. BLLT'20 [3] show that their ``benign'' condition is *almost* necessary for expected risk consistency. Essentially, looking at the combination of BLLT'20's lower bound and NDR'20's upper bound the ``weakly benign'' condition of NDR'20 appears to be necessary and sufficient for risk consistency. How do the results of the present paper, at least for $\hat w_{MN}}$ compare to those results? 3. The role of 1-sided v.s. 2-sided bounds remains unclear, though discussion of 1-sided bounds possibly holding while 2-sided bounds fail makes up a fairly significant portion of Sec.~3.2 . The results of Sec.~4, which involve ``restricted'' uniform bounds (the sup is constrained to predictors which interpolate) are stated as 1-sided results, but clearly hold as 2-sided results since $L_S=0$ by the constraint and $L_D\ge 0$. It seems that ``restricted'' uniform bounds will always have this property. It may be useful to understand the role of 1-sided vs 2-sided bounds in relation to Sec.~4, or to briefly explain why the methods of Sec.~4 circumvent this issue. POST REBUTTAL COMMENTS: *Portable Insights*: Further discussion of optimistic rates, and the possibility of applying duality in subsequent work would be helpful. It is not obvious to me how the duality argument would work for problems other than linear regression. *Comparison to [3]/[23]*: Maybe it would be helpful to show that bounds on the restricted eigenvalue that lead to consistency imply the "bening" condition would hold, and intuitive relationships between them would be helpful. Probably in an appendix would be good. I agree that the restricted eigenvalue and the norm of X are more easily interpreted than the quantities appearing in the "benign" condition.

Correctness: 1. The claims all seem to be correct.

Clarity: 1. The paper is readable. 2. There is quite a bit of math interspersed with the discussions, making some parts hard to follow. Some results are stated and argued to be correct ``conversationally''. Thee could instead be lemmas/propositions/.... For example the itemized results immediately before Prop~2.1 could be stated as a formal lemma/prop/... 3. The abstract could more concretely state their results. There is a question raised in the abstract that I'm not sure is definitively answered in the paper; ``is it low norm that is key to good generalization, rather than some other special property of this particular solution?'' If this is going to be framed as a main motivating question of the paper, it would be good to clearly state the answer to it. The answer could even appear in the abstract 4. Some parts could be rewritten to have simpler sentence structures and shorter sentences. E.g. Lines 230-232 were particularly hard to follow for me, and I had to reread it several times. POST REBUTTAL COMMENTS: *Is low norm key?*: It seems that low norm is sufficient, but may not be necessary unless I am missing something.

Relation to Prior Work: 1. The appropriate related work is cited, as far as I can tell. 2. Some novel aspects of this work are clearly stated (e.g. the extension to ``small''-norm ERMs 3. some relationships between results and methods in this work could be better juxtaposed relative to existing work. 3.a) (as per weaknesses 2.) The comparison of the sufficient condition for consistency with the necessary and sufficient conditions of BLLT'20+NDR'20 is missing 3.b) (as per additional feedback 1.) It is not clear if the ``restricted'' uniform bound (e.g. Eq.~(7)) appeared before in the literature.

Reproducibility: Yes

Additional Feedback: 1. Has the ``restricted'' uniform bound (e.g. Eq.~(7)) appeared before in the literature? Is there a general reason we might expect terms like this to be small where their unrestricted counterparts are not? Is that the idea in this work that is the most ``reusable'' in to obtain positive results for other applications? If so you should highlight this earlier and more clearly so that it stands out to future readers. 2. Footnote 3 could be justified with a simple application of DCT, as far as I can tell. Right now it refers the reader to the "stronger" result of Prop.~4.6, which is only for the limit as $n\to \infty$. 3. Thm.~4.5 is missing a ^2 on $\norm{...}$ 4. My initial score, a 6, is based on the current state of the manuscript. If the manuscript changed by exactly addressing the points I've raised above (or they were addressed adequately in the rebuttal), I believe I would raise my score to a 7 based on the novelty and scope of the results. In order to go higher than 7, I think I would need to be shown that there is a substantial additional contribution which I've failed to acknowledge or the content of the manuscript would need to change in some significant way. POST REBUTTAL COMMENTS: I'm on the fence about whether to change my score to a 7 or not. The 6/7ness of my score would be entirely based on the quality of the discussion added by the authors based on the *Portable insights* item under the "weaknesses" section of my review. Because of the lack of detail in the response to this item, and since I cannot actually review the content that they will add, I have erred on the side of conservatism and not increased my score. Based on the other scores, it seems like the paper is likely to be accepted anyway. I still hope the authors will provide a good discussion around the reusability of the methods, both for the longer-term impact of their work, and to help the community apply similar techniques to other problems if the techniques prove more broadly applicable.


Review 4

Summary and Contributions: In this work, the authors show that uniform convergence can be used to prove consistency for interpolation learning given a linear regression example.

Strengths: The paper gives a proof about how to use uniform convergence to prove consistency for a low-norm interpolation learning problem.

Weaknesses: The paper mentions deep learning. But it is not clear what is the relation of this work to deep learning. The paper goes through the proof step by step. But it lacks discussion about the importance of connecting uniform convergence to interpolation learning. The proof is also for a simple high-dimensional linear regression and this assumption seems to be strong.

Correctness: The proof seems to be valid.

Clarity: The paper goes through the proof step by step. Because of that, the paper lacks discussion about the importance of the work, e.g., why we want to connect uniform convergence with interpolation learning and how a proof on linear regression can justify the claim is still true for nonlinear cases. The authors mentioned deep learning then the theory is expected to go beyond linear cases.

Relation to Prior Work: The paper mentions that the problem in this paper is proved but not using uniform convergence to prove it. The paper lacks discussion to make the claim solid about why we have to use uniform convergence to prove the theory. Also, for Lasso regression, it can use uniform convergence bound as a starting point to prove the consistency of the parameter. It seems that the authors need to include more statistics papers in the literature review.

Reproducibility: No

Additional Feedback: Thanks for the feedback. I decide to change my score to 'accept' for the value in this paper to rethink the role of uniform convergence.

[Author Response · NeurIPS 2020]

**Reviewer #1**  Thanks for your comments; we'll clarify our usage of the following terms in the revised paper.

• We use **"consistency"** to mean $\mathbb{E}[L_{\mathcal{D}}(\hat{w}) - L_{\mathcal{D}}(w^*)] \to 0$. Traditionally, as in [27], this limit means $n \to \infty$ for a
fixed problem, but in that setting linear models do not interpolate. Instead, for asymptotic interpolation we study
a sequence of distributions changing with $n$, with the noise magnitude $\lambda_n$ possibly increasing. In a more typical
"high-dimensional" regime, $p$ would also increase with $n$, e.g. $p = \gamma n$ in [13]; we instead take $p \to \infty$ for each $n$.
• By **"interpolation learning"** we mean achieving "good" $L_{\mathcal{D}}(w)$ while $L_S(w) = 0$ in a noisy, non-realizable setting.

**Reviewer #2**  Thanks for your feedback; we'll add more intuition, details, and reorganize proofs in revision.

• **Min-risk interpolator:** Thm. 4.5 decomposes as risk of one interpolator, plus gap to worst; $\hat{w}_{MR}$ minimizes risk.
• **Restricted eigenvalue:** It arises naturally from (7)'s dual; it measures how of $\Sigma$ is unobserved by $X$, and is the
generalization gap for $y = 0_n$, $B = 1$. It also relates to [3]: the "malignant" covariance $I_p$ has $\kappa_X(I_p) \stackrel{a.s.}{=} 1$, while
the benign covariance of Setting B has $\kappa_X(\Sigma) \approx \lambda_n/n \to 0$. We expect it might play the role of $\xi_n$ in $(\star)$.
• **Finite degrees of freedom:** It is true that Setting B is simple in this way. Our approach also allows for analysis
where $d_S$ increases with $n$, though we know it must be $o(n)$ for consistency to be possible.
• **Consistency $\to$ 1-sided unif. conv.:** Take $\mathcal{S}_{n,\delta} = \{(X,y) : L_{\mathcal{D}}(A(X,y)) \le \sigma^2 + \epsilon_{n,\delta}\}$. (We'll clarify footnote 5.)

**Reviewer #3**  Thanks for your writing suggestions (converting some discussion into lemmas, substantially re-focusing
the abstract, and clarifying e.g. line 230), which we agree will improve the presentation.

• **Portable insights:** The main takeaway we believe to be broadly relevant is that when analyzing using "uniform
convergence," especially in the context of interpolation learning, it is important to use "relative" or "optimistic"
bounds which take $L_S$ into account. Our approach of bounding the generalization gap via duality may also be widely
applicable: even in complex settings without strong duality, upper bounds should still be possible from weak duality.
We will emphasize these more throughout the paper.
• **Comparison to [3]/[23]:** While prior work almost fully characterizes consistency in this class of problems, it is
quite different from most existing work in statistical learning theory. Our theorem 4.5 attempts to be more like
popular Rademacher bounds, although to develop this connection further (and compare with existing conditions),
more calculations are required in general – even if the speculative bound $(\star)$ holds. We'll increase our discussion of
the relationship to the benign/weakly benign conditions, e.g. with the examples above. Our approach also explains
non-minimal-norm predictors, and it may be easier to numerically check $\kappa_X(\Sigma)$ and $\|X\|$ in practice.
• **1- vs 2-sided uniform convergence:** For predictors with $L_S(w) = 0$, these modes of convergence are indeed
identical. These restricted uniform bounds sidestep entirely the two-sided failure mode of Section 3.2, with high $L_S$
but low $L_{\mathcal{D}}$. This is not the only difference between the standard and restricted settings, however: we strongly expect
that norm balls do not exhibit one-sided uniform convergence either (line 125), due to cases where $L_S$ is large but
$L_{\mathcal{D}}$ is even larger. We will add more discussion of this relationship in the revision.
• **Restricted eigenvalue under interpolation:** We are not aware of any previous use of $\kappa_X(\Sigma)$ in the literature.
• **Is low norm key?** As any low-norm interpolator generalizes, we believe we've shown that the answer to this question
is "yes." We agree that this belongs in the abstract and should be highlighted more in the paper body as well.
• **Restricted convergence bounds:** As we mention around line 177, bounds like (7) are very standard in realizable
PAC analyses. Generally, (7) will *always* be small for consistent predictors – even if, as in Section 3, unrestricted
bounds fail – because taking $B = \|\hat{w}_{MN}\|$ makes (7) just $L_{\mathcal{D}}(\hat{w}_{MN})$. The questions are whether we can usefully
bound the analogue of (7), and how large $B$ can be; we answer these questions for Setting B in Section 4.1.
• **DCT in footnote 3:** Since $L_{\mathcal{D}}$ is also an expectation, there are two interchanges of limit and expectation, and finding
a dominator seems nontrivial; it seems to essentially boil down to the proof of Proposition 4.6.

**Reviewer #4**  Thank you for your questions and suggestions, which we will emphasize in revision.

• **Connection to deep learning:** High-dimensional linear models can serve as a simpler test bed to help develop
methods useful for deep nets: we need to walk before we can run. Knowing which techniques can explain many of
the surprising phenomena of deep learning, e.g. double descent, in linear models, helps us narrow down which tools
to try in the harder setting. (See also the *portable insights* comment to Reviewer 3.)
• **Why uniform convergence?** (1) It is in many ways *the* standard toolkit in statistical learning theory. (2) A direct
bound on $L_{\mathcal{D}}(\hat{w}_{MN})$ may not tell us *why* $\hat{w}_{MN}$ works; a uniform bound based on norm strongly indicates norm is the
"reason." (3) In practice we may not find the exactly minimal-norm interpolator; uniform bounds are more "robust."
• **Restricted problem setting:** Indeed, Theorem 4.1 is limited to a very particular setting, but we use it mainly to
demonstrate the success of our style of analysis. We emphasize that Theorem 4.5 holds quite generally.
• **Comparison to LASSO:** Here, we simply make the point that in a sparse setting, there exists a consistent learning
rule, but *no* interpolation method – including the minimal $l_1$ norm interpolator – can be consistent for $p = \mathcal{O}(n)$.
• **Related statistics papers:** If you have any particular work in mind, we are eager to consider it.

[Meta-Review · NeurIPS 2020]

The paper considers over-parameteried linear regression, explores uniform convergence to see if consistent minimum norm solution can be recovered, illustrates that UC does not help here, but focusing on a restricted setting (interpolation) holds promise. The reviewers appreciated the main results. It is unclear if the analysis and results has implications beyond the linear setting considered.